# Were climatic forcings the main driver for mid-holocene changes in settlement dynamics on the Varamin Plain (Central Iranian Plateau)?

**Fabian Kirsten** [1⊙]*, **Anne Dallmeyer** [2⊙]*, **Reinhard Bernbeck**[3], **Thomas Böhmer**[4], **Robert Busch**[1], **Morteza Hessari**[5], **Susan Pollock**[3], **Brigitta Schütt**[1]

**1** Freie Universität Berlin, Division of Physical Geography, Berlin, Germany, **2** Max Planck Institute for Meteorology, Hamburg, Germany, **3** Freie Universität Berlin, Institute of Near Eastern Archaeology, Berlin, Germany, **4** Alfred-Wegener-Institute for Polar and Marine Research (AWI), Postdam, Germany, **5** Cultural Heritage and Tourism Research Institute, Tehran, Iran

⊙ These authors contributed equally to this work.
* fabian.kirsten@fu-berlin.de (FK); anne.dallmeyer@mpimet.mpg.de (AD)

**Data Availability Statement:** The speleothem-based climate reconstructions can be downloaded from the NOAA Database (https://www.ncei.noaa.

## Abstract

Settlement crises in ancient cultures of Western Asia are commonly thought to be caused by climatic events such as severe droughts. However, the insufficient climate proxy situation in this region challenges the inference of clear relationships between climate and settlement dynamics. We investigate the Holocene climatic changes on the Varamin Plain in the context of the climatic history of Western Central Asia by using a transient comprehensive Earth System Model simulation (8 ka BP to pre-industrial), a high-resolution regional snapshot simulation and a synthesis of pollen-based climate reconstructions. In line with the reconstructions, the models reveal only slightly varying mean climatic conditions on the Varamin Plain but indicate substantial changes in seasonality during the Holocene. Increased precipitation during spring, combined with lower temperature and potentially stronger snow accumulation on the upstream Alborz mountains may have led to an increased water supply on the alluvial fan during the vegetation period and thus to more favourable conditions for agricultural production during the Mid-Holocene compared to modern times. According to the model, dry periods on the Central Iranian Plateau are related to particularly weak Westerly winds, fostering the subsidence in the mid-troposphere and hampering precipitation over the region. The model reveals that dry periods have spatially heterogenous manifestations, thus explaining why they do not appear in all proxy records in the wider study region. In fact, the climatic signal may depend on local environmental conditions. The interaction of the topography with the atmospheric circulation leads to additional spatial heterogeneity. Although our results provide several indications for a connection between climate and settlement dynamics, the small overall changes in moisture call into question whether climate is the main driver for settlement discontinuities on the Central Iranian Plateau. To shed further light on this issue, more high-resolution long-term proxy records are needed.

gov/access/paleo-search/). All other datasets and analyse scripts used for the analysis and plots in this study are deposited in the MPG publication repository: https://hdl.handle.net/21.11116/0000-000D-14BA-B and can be downloaded free of charge. In addition, a preliminary version of the pollen-based climate reconstruction (LegacyClimate 1.0) for the Northern hemispheric extratropics is provided as open-access data on PANGAEA (https://doi.pangaea.de/10.1594/PANGAEA.930512). Further variables of the high-resolution snapshot simulation can be downloaded from the long term archive of the German Climate Computing Center (DKRZ), accredited as regular member of the World Data System (https://doi.org/10.26050/WDCC/ICON-NWP_mH_pd).

**Funding:** This study contributes to the project "Mobile villages and dynamic landscapes: the Varamin Plain from the late 5th to the early 3rd mill. BCE" (Project number 424609853) within the priority programme 2176 "The Iranian Highlands: Resilience and Integration of Premodern Societies" funded by the Deutsche Forschungsgemeinschaft (DFG, German Research Foundation, www.dfg.de). (FK, RBe, SP, RBu, BS) This work contributes to the project PalMod, funded by the German Federal Ministry of Education and Research (BMBF, www.bmbf.de), Research for Sustainability initiative (FONA, www.fona.de) and the Cluster of Excellence EXC 2037 "CLICCS - Climate, Climatic Change, and Society" (Project Number: 390683824), funded by the Deutsche Forschungsgemeinschaft (DFG, German Research Foundation, www.dfg.de) under Germany's Excellence Strategy. AD was financed by PalMod (Grant number: 01LP1920A) and partly by CLICCS. The funders had no role in study design, data collection and analysis, decision to publish, or preparation of the manuscript.

**Competing interests:** The authors have declared that no competing interests exist.

# 1. Introduction

The relationship between the dynamics of cultural developments, settlement dynamics and periods of crisis has been in the center of scientific attention with respect to regions harbouring the early development of sedentary societies and archaeological remains in Western Central Asia (WCA). So far, the majority of these investigations have focused on Western Asia in general and Mesopotamia in particular [1–3]. Within this context, there is an ongoing debate regarding the reasons for settlement discontinuities based on archaeological findings throughout WCA and specifically for the area of modern-day Iran and the Central Iranian Plateau (CIP) [4–7].

As one possible driver for decreases in settlement density and intensity, climatic events, mainly in the form of dry periods, are discussed. Based on different proxy records, several climatic events have been identified for the Holocene period on a regional scale [6, 8, 9]. This includes severe droughts or cold spells attributed to the time periods around 8.2, 5.9, 5.2 [10, 11], 4.2 and 2.6 (Assyrian megadrought) ka BP [12–17].

Some scholars have proposed that these events may even have caused the demise of various cultures, such as the Late Uruk period societies in Mesopotamia (5.2 ka) [18, 19] or the collapse of the Akkadian Empire in Mesopotamia, the Old Kingdom in Egypt, or the Lianhzhu culture in China (4.2 ka, e.g. [20–22]).

In particular, climatic changes have been discussed as a major reason for the striking disappearance of most settlements on the CIP after the Proto-Elamite Period/Early Bronze Age (EBA) at roughly 5 ka BP. This conclusion has been derived from the absence of archaeological sites for the centuries thereafter [17, 23, 24]. Settlement density increases again during the Late Bronze or Early Iron Age (around 3.5 ka BP) [4, 17]. An arid period between 4.55 and 3.25 ka BP termed "Central Iranian Drought" [25], often also associated with the 4.2 ka climate event [17], is commonly identified as a main driver of settlement discontinuities. This is in line with the results of a recent (meta)study by Palmisano et al. [26] indicating a more or less continuous decline in settlement density for the region of modern-day Iran from about 4.7 ka BP onwards, following an approach often labelled "dates as data" (i.e. numbers of dated samples are assumed to correlate with data points for settlement density, etc.) [27]. In this regard, the 3rd millennium BCE (5–4 ka BP) is considered as a key transition period in terms of climatic conditions [28, 29] and the simultaneous emergence of more urban societies on the CIP [7, 30].

Similar temporal settlement patterns and discontinuities have been identified for the Varamin Plain (VP), an alluvial fan east of Tehran (Iran), based on a survey conducted in 2010/2011 by M. Hessari [24, 31]. Alluvial fans have long been identified as preferred locations for early settlements and the development of agriculture [32] due to their additional water supply by runoff. The VP and other alluvial fans on the southern slopes of the Alborz mountains have been studied intensely in the last decades with respect to geoarchaeological questions [2, 25, 33, 34]. In the further course of archaeological investigations on the VP in the context of the research project "Mobile villages and dynamic landscapes: the Varamin Plain from the late 5th to the early 3rd mill. BCE", questions arose regarding possible climatic explanations for the identified periods of crisis during the Mid-Holocene [35]. However, continuous proxy records for the entire Holocene do not exist for the VP and are generally scarce on the CIP. Additionally, the available reconstructions are in most cases derived from speleothems due to the absence of lake records (except for the area of the Zagros mountains in western Iran).

Paleoclimate modelling can close this gap and can give valuable insight into the climatic past, particularly in regions with low proxy record density. For the area of modern-day Iran, [36] could show that winter rainfall patterns changed towards wetter conditions starting around 3 ka BP due to changes in insolation and a resulting southward shift in the West Asian

Subtropical Westerly Jet. The Westerly Jet is mainly responsible for moisture transport to Central Iran including VP during the rainy period in winter and spring [37]. A multi-model inter-comparison study (PMIP3/PMIP4) indicates dryer conditions in (mainly eastern) Arid Central Asia (ACA) for the simulated mid-Holocene time-slices (6 ka) compared to the simulated pre-modern (pre-industrial) climate [38, 39]. This aridification trend is mainly caused by a reduction in spring precipitation due to weakened Westerlies and reduced atmospheric water vapor content. This contrasts with findings based on proxy records that discuss an increased monsoonal influence, including summer rainfall and generally moister conditions for Early- to Mid-Holocene times for areas in and around the Zagros mountains and on the CIP [3, 40]. A consistent comparison of climate model simulations and proxy records for the western part of Central Asia has—to our knowledge—not been conducted so far. In particular, due to the spatial resolution of these climate models and their global nature, they have not been applied in archaeological contexts on a local to regional scale for the CIP.

Recently, transient paleoclimate simulations for the last 8000 years in a relatively high spatial-resolution have been conducted at the Max-Planck-Institute for Meteorology Earth System model MPI-ESM1.2. These simulations have already been successfully evaluated with respect to Holocene temperature changes [41], global vegetation change [42] and the end of the African Humid Period [43]. In this study, we explore one of these simulations regarding the Holocene climate trend in WCA with particular focus on the CIP and VP. By additionally analysing archaeological survey data from the VP, openly available proxy data, proxy-derived climatic reconstructions within the LegacyClimate 2.0-framework (based on the LegacyClimate1.0 dataset by [44]) and a high resolution snapshot simulation by ICON-NWP, we focus on the following overarching research questions:

1. How do circulation patterns and the resulting broad-scale climate evolve over the course of the Holocene in WCA/CIP? How did climatic seasonality change?

2. Does the model output correspond to the existing information from terrestrial archives/proxies (pollen, δ18O) regarding the general climatic trends? What can we learn from the model about aridity events commonly discussed as possible explanation for settlement crises in archaeological studies covering the Mid-Holocene [6, 9, 30]?

3. To what extent does information about paleoclimatic conditions derived from proxies and models correspond to archaeological evidence on different spatial and temporal scales?

## 2. Study area

This study comprises three different spatial scales Fig 1. Our central (archaeological) case study relates to the VP on the Jajrud alluvial fan east of present day Tehran Fig 2. This study area is archaeologically but also climatically embedded in a larger research area, which roughly corresponds to the CIP. Thirdly, based on the distribution of regional archives and paleoclimatic proxies as well as the spatial resolution of the model data, we analyse model and proxy data for a larger study area that corresponds roughly to the region labelled Western Central Asia (WCA) by the Intergovernmental Panel on Climate Change (IPCC) (including the easternmost parts of the MED-area as defined by the IPCC) [45] Fig 1.

### 2.1. Geomorphological setting

The "Central Iranian Plateau" is defined differently depending on the scientific context. Generally, it denominates the southernmost part of the Eurasian Plate (i.e. roughly the central part of modern day Iran) and is bordered by the Zagros Mountains in the west and the Alborz

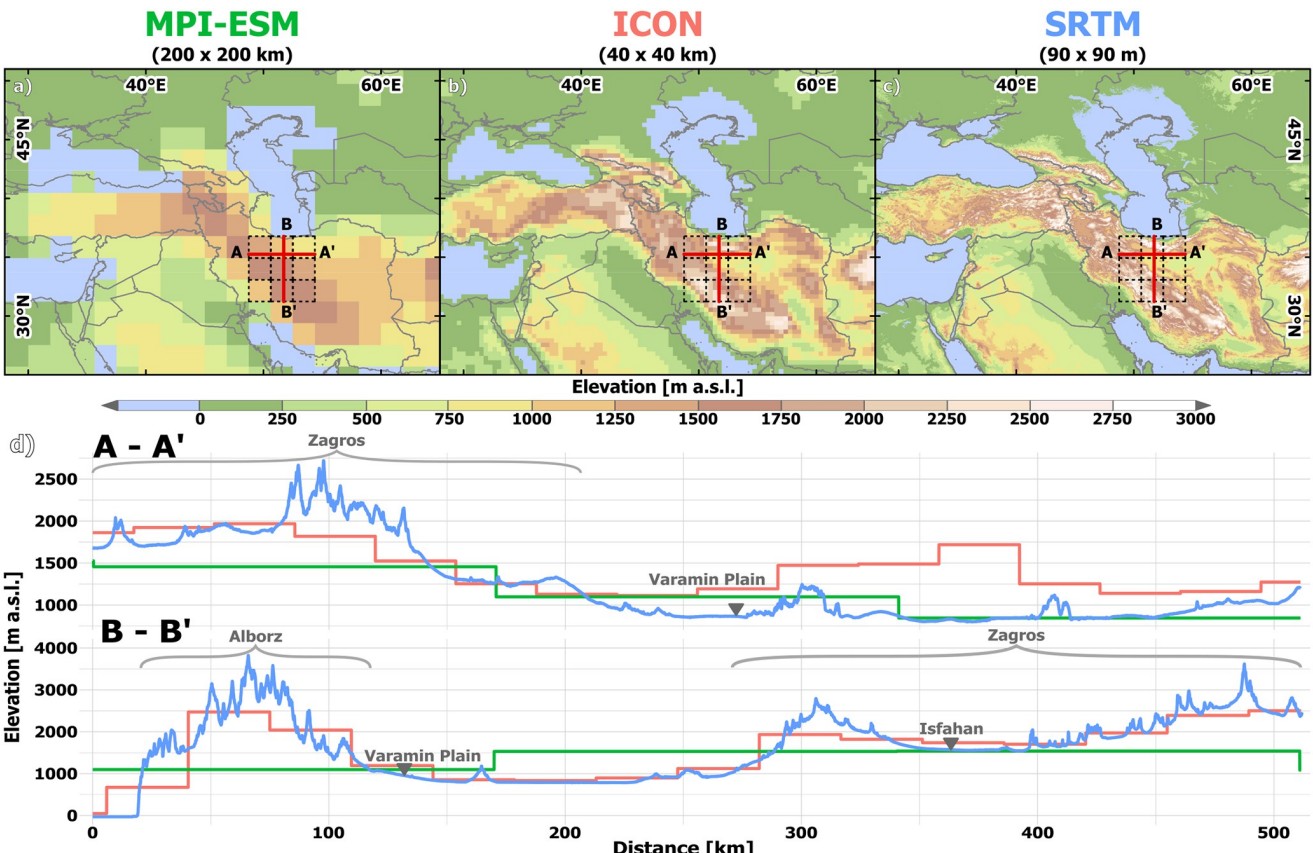

**Fig 1. Orography of the wider study area (Western Central Asia) in different model resolutions.** Location of the study area (the Varamin Plain is located at the intersect of the elevation profiles A-A' and B-B'. The 9 grid cells comprising the Central Iranian Plateau in the MPI-ESM model are marked by dashed lines) before the background of the orography of (a) MPI-ESM on a T63 grid. (b) ICON NWP (mapped on a 40 x 40 km grid). (c) SRTM 90 [46]. (d) East-West-profile A-A' and North-South profile B-B' visualising the differences in orography between the MPI-ESM resolution (green line), the ICON NWP resolution (red line) and a modern DEM (SRTM) (blue line). The country and ocean boundaries are based on the feature layer "Global_Ocean_Country_Masks" (ID: 91) in ArcGIS Online.

mountain chain in the north. Despite the term plateau, the area contains several mountain ranges and even areas below sea level. In the context of this study, we apply the term for the larger study area of about 600 x 600 km around the VP that serves as a point of comparison for regional climatic variability within the model data Sec 3.2.

The Varamin Plain (sometimes included under the label Tehran Plain [34]) is a piedmont basin on the norhtern edge of the CIP adjacent to the Alborz Mountains and covered with fluvial, gravelly to sandy/silty sediments of the alluvial fan of the Jajrud river. Due to the continued tectonic uplift of the Alborz Mountains, denudation of the orogen ensures a high supply of sediments with changing regimes of erosion and sedimentation and a constant flux in channel genesis on the alluvial fan [33]. The thickness of alluvial sediments on the VP ranges between 100 and 200 m, in certain areas of the fan even up to 300 m [47]. Due to the coarse (sandy) nature of these Quaternary fan sediments, they contain several aquifers that constitute an important water resource in the study area that has been made accessible through the construction of numerous qanats [48].

The time frame for the genesis of the alluvial fans of the Alborz mountains is not very well investigated. While the late Pleistocene and Early Holocene were probably rather dry with

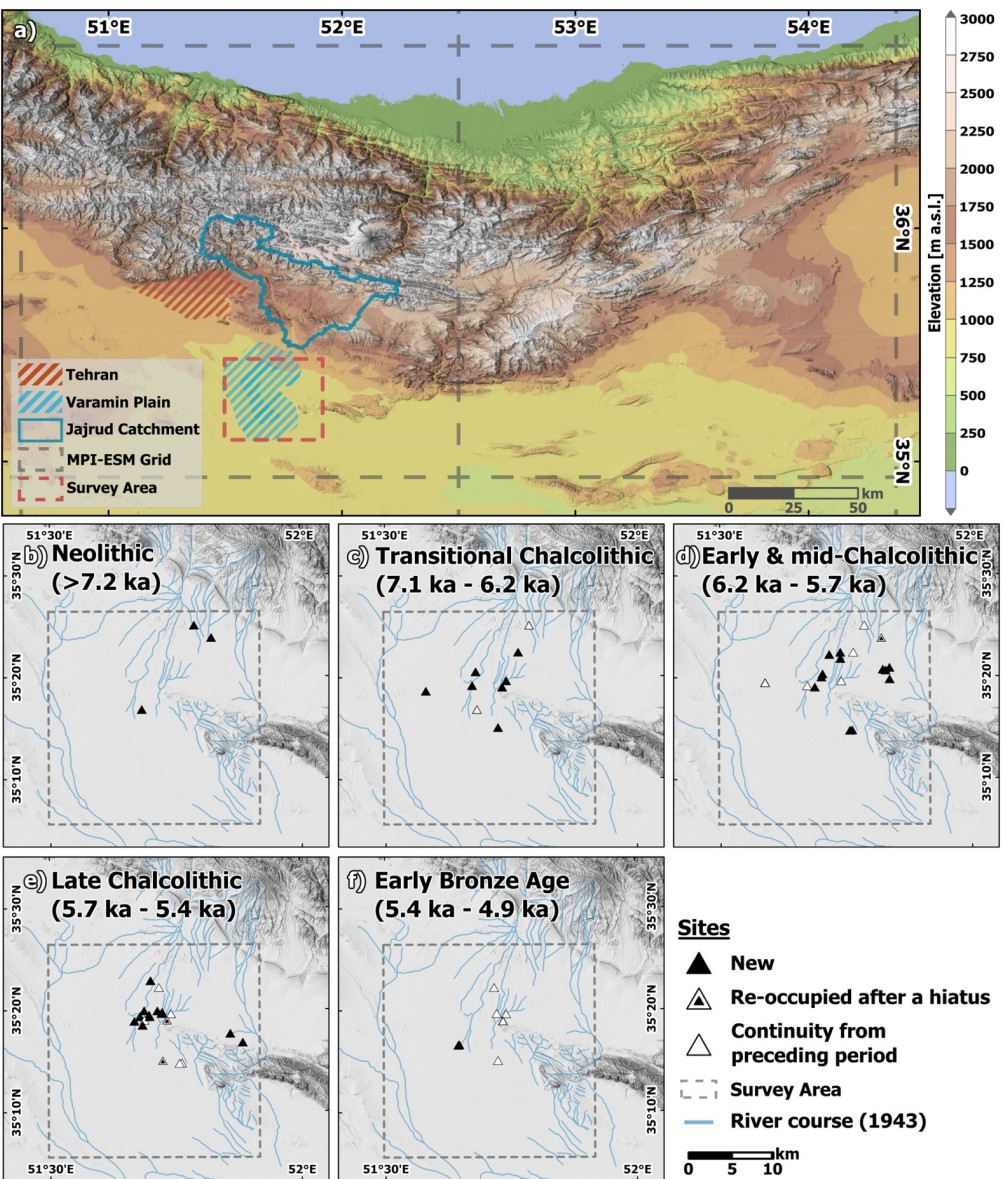

**Fig 2. Location of the study area and archaeological survey data for the Varamin Plain.** (a) Location of the survey area in the regional topography and in relation to the MPI ESM grid. (b-f) Archaeological survey area on the Varamin Plain with location of sites with archaeological remains during respective archaeological periods (DEM: SRTM 90 [46]; River courses are based on the U.S. Army Map Series K501: Iraq and Iran, Tehran I-39D, 1943); Note: The grid cell in the MPI-ESM model used for subsequent analysis in this study is the grid cell adjacent (east) to the grid cell containing the VP (see method section for details).

sheetflow-style sedimentation, moister conditions associated with increased runoff and channel flow going along with channel incision prevailed after ca. 7.2 ka BP. For the Mid-Holocene, a steady but slow reduction in moisture until 4.55 ka BP [25] or even 3 ka BP [33] has been attested. This aridification is inferred from an absence of younger fluvial deposits on the alluvial fans along the northern edge of the CIP. Moister climatic conditions associated with sedimentation, partially covering the Mid-Holocene sediments, and even soil formation were identified by Gillmore et al. [33] for the Late Holocene and confirmed by findings of Büdel

[49] on the nearby Damghan alluvial fan. Fluvial sediments at the archaeological site of Ajor Pazi a few kilometers south of the modern town of Varamin on the VP were dated to ages between 8 and 6 ka BP by radiocarbon and luminescence dating [35].

The watershed of the Jajrud River has an extent of about 1892 km$^2$ with its headwater area being located in the Alborz mountains [47] Fig 2. At the entrance into the VP (apex at appr. 1160 m a.s.l.) [50], the river branches into several channels, of which only few are perennial. The main discharge occurs in the months from April to June fed by snow melt in the Alborz mountains and spring precipitation.

## 2.2. Climatic setting

The climate in modern day Iran is continental with an annual temperature range of up to 26˚C [51]. Summers are hot and dry and characterised by subtropical highs. Winters can get very cold [37]. The rainy season lasts from November to May in most parts of the country. The rainfall pattern is mainly influenced by mid-latitude Westerlies, embedding Mediterranean synoptic systems that are responsible for most of the precipitation in the region. The number of passing cyclones per year varies strongly. This leads to a high interannual and interdecadal rainfall variability [52]. The Zagros and Alborz mountain ranges act as barrier for the atmospheric flow, preventing most rainfall-bearing systems to reach further inland to the central and eastern parts of the country. Therefore rainfall concentrates particularly on the windward slopes of the mountains [53] while the eastern part of modern day Iran experiences a desert climate with annual precipitation rates of less than 100 mm. Overall, the climate is arid to semi-arid. Mean precipitation across the country sums up to approx. 240 mm/a [51]. The complex topography, thus, plays an important role for the regional heterogeneity in rainfall.

The VP is located in the drier part of the CIP. In addition to Mediterranean cyclones, the northern part of Iran including the VP also receives considerable amounts of air moisture from the Caspian Sea (as well as from the Black Sea and the Persian Gulf) [37]. The modern mean annual precipitation on the Jajrud alluvial fan amounts to about 140 mm [47] with high interannual variability and a rainy season from December to April [47]. For the nearest official weather station at Tehran Mehrabad, which is located several hundred meters higher than the VP, annual precipitation amounts to about 260 mm and mean annual temperature is 18.2˚ C Fig 3. On the Jajrud watershed in higher altitudes of the Alborz mountains, precipitation reaches annual amounts of over 1000 mm. The potential evaporation on the VP amounts to approx. 2000 mm a$^{-1}$ with a pronounced water deficit in spring, summer and fall [47].

## 2.3. Archaeological setting

Archaeological fieldwork in combination with modern techniques of absolute dating have allowed for a relatively precise chronological framework for prehistoric societies on the CIP [7, 24, 54, 55] Tbl 1. The earliest traces of occupation on the CIP date back to the Late Paleolithic, with substantial indications of settlement dating to the Late Neolithic [56]. The VP contains only few settlement traces from the Late Neolithic and notably more settlements during the Transitional Chalcolithic, even if the overall number remains low [24]. An increasing density of settlements on the CIP is observable for the Early/Middle Chalcolithic (appr. 7.3 to 6.3 ka BP) and Late Chalcolithic periods (6.3 to 5.5 ka BP) [7, 23, 57], also attested in the VP [24]. Of particular interest are questions related to the apparent similarities in material culture across a larger region in a period labeled "Proto-Elamite", characterized by a sharp drop in site numbers and a material culture that is starkly different from earlier periods. Administrative procedures, and particularly an emerging early writing, introduced from lowland Khuzestan, seem to have played a prominent role, although the socio-economic importance of these mnemonic

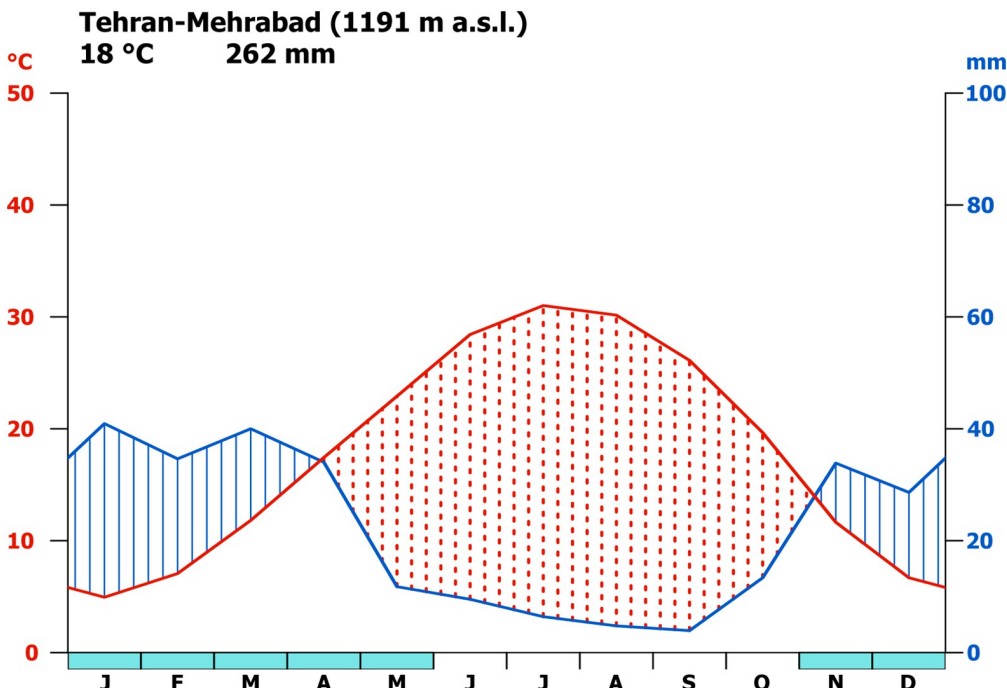

**Fig 3. Climate diagramm of station at Tehran-Mehrabad.** Climate between 1991–2020 at Tehran-Mehrabad (35.683 N, 51.317 E) located at 1191 m a.s.l. (raw data downloaded from https://www.ncei.noaa.gov/cdo-web, last access May 9th 2023).

technologies for societies in the highlands remains unclear. This cultural phase is dated to the Early Bronze Age (Sialk IV,1 and IV,2). Uncertainties in dating remain due to methodological difficulties related to a plateau in the radiocarbon calibration curve between 5.4 and 4.9 ka BP [58]. There is convincing evidence from the northern Central Plateau/CIP that the location of sites followed the occurrence of springs, (perennial) river channels, and communication routes [23].

At the end of this period, mobility and settlement shifts seem to have led to an abandonment of the VP for several centuries (approximately 4.9 to 4.1 ka BP) [24, 59], while the period of crisis probably lasted well into the 2nd millennium BCE (4 ka BP or later). This is often interpreted as a shift from a sedentary to a nomadic way of life [56], while the potential reasons for these changes remain unclear. Considering these long-term social developments on the CIP, the periods from the Late Neolithic to the Early Bronze Age II are a focus of current archaeological research on the VP [31].

While paleoclimatic shifts are often explicitly or implicitly implied, recent archaeological studies on the Varzaneh plain in the interior of modern-day Iran have shed light on flourishing settlements during this Early Bronze Age II (4.8–4.2 ka BP) [17], as well as further southeast in the region of Jiroft and Konar Sandal [60]. These developments are in direct contrast to the findings from the VP and other areas on the CIP. The changes in settlement or population densities (as a proxy for cultural stability and development) on the VP are deduced from survey results of the number of archaeological sites for each respective period during a survey carried out by M. Hessari on the VP in 2010/2011 Fig 2. For the northwestern regions of the CIP, a dispersed set of small rural sites has been observed, while in the south, rural abandonment, in combination with the growth of single, more densely populated regional centres is attested [7].

# 3. Materials and methods

## 3.1. Archaeological data

Data from an archaeological survey of the Jajrud alluvial fan/VP by Morteza Hessari [24, 61] was used for the identification of spatial and temporal settlement patterns Fig 2. The timing of archaeological periods has been assigned based on the analysis by Pollard et al. [54] (cf. Tbl 1). A higher temporal resolution of the dataset cannot be achieved due to the nature of archaeological finds and the scarcity of corresponding absolute dates derived from 14C-dating. Furthermore, different terminologies and chronologies apply to sub-regions in the wider study area (CIP, WCA), further complicating intra- and inter-regional comparisons [16].

## 3.2. Model data

**3.2.1. Transient simulation in MPI-ESM.** We use a transient simulation of past climate for the last ~ 8000 years (6000 BCE to 1850 CE) that has been conducted in the Max-Planck-Institute Earth System Model 1.2 (MPI-ESM) [62]. This is a comprehensive Earth System Model, including interactive coupling of the atmosphere, ocean and land surface and vegetation. The atmosphere and land model work in a spatial resolution of T63L47, which are approx. 200 km per grid cell on a Gaussian grid, and 47 levels vertically. The model is forced with changes in orbital-induced insolation, greenhouse gas concentration, stratospheric aerosol injections imitating volcanic eruptions, and spectral solar irradiance changes (e.g. solar cycle). For the last 1000 model years, land-use has been additionally prescribed, with a linear transition period of another 1000 years for building it up. A detailed description of the transient simulation and its forcings is given in [42].

Although the model ran in a relatively high resolution for paleoclimatic simulations, the limited representation of the detailed topography of WCA in this resolution affects the simulated regional climate. The Zagros mountains are rather flat, the Alborz mountains are not represented, and the VP is located on the eastern flank of the Zagros in this resolution Fig 1. For the study area representing the CIP, a domain of 3x3 cells was used. As a spatial representation for the archaeological survey area on the VP, the north-eastern cell of these 9 grid cells was chosen (center: 52.5 E, 36.37 N) Fig 1. With a mean elevation of 846 meters in the model, this grid cell closely corresponds to the real elevation on the VP. In addition, the simulated pre-industrial climate resembles the modern observations.

Given that climatic changes on the VP are a result of changes in the large-scale atmospheric circulation patterns, we are certain that this simulation is appropriate for analysing the climate variability on the VP during the Holocene. This simulation and a slightly different one in the same model have already been successfully used to shed light on prominent climatic changes in the Holocene such as the end of the African humid period [43] or the Holocene temperature conundrum [41].

The model is able to simulate temporal variability on different time scales that can be related to the prescribed forcing or to the internal variability of the simulated climate system. Thus, single events caused by factors beyond the given forcings cannot be reproduced by the model. Similarly, climatic events that are related to internal variability may not occur at the same time in the simulation as discovered in reality.

**3.2.2. The high-resolution snapshot simulation in ICON-NWP.** For a spatially more detailed analysis, the results of a snapshot simulation for 7 ka BP with a spatial resolution of 40x40 km Fig 1 performed by Jungandreas et al. [63] is used. This simulation has been performed in the model ICON-NWP version 2.5.0 [64], which is a nonhydrostatic model and commonly used as a standard model framework in the German Weather Forecasting Service.

The simulation has been forced by the initial and lateral boundary conditions from a transient global Holocene simulation by MPI-ESM, that differ slightly from the MPI-ESM simulation used in this study. Land-surface parameter have been prescribed from reanalysis data of the Integrated Forecast System (IFS) of the European Centre for Medium-Range Weather Forecasts (ECMWF). In this simulation, they have been adapted to the Mid-Holocene vegetation cover simulated by MPI-ESM, to represent, e.g., the Green Sahara. In this high-resolution regional simulation, the orography and therewith its influence on the regional circulation is assumed to be captured more accurately than in the global model. However, the complex orography of the Zagros and Alborz mountain ranges is still not fully represented in this resolution and—since the regional simulation is nested into the global simulation–possible biases in the simulated large-scale circulation in MPI-ESM also directly affect the circulation in the regional model.

As the representative grid-cell for the VP in the ICON-NWP model, we choose the grid-cell with the center at 51.75 E, 35.40 N with a mean elevation of 1193 m. The Jajrud catchment area in the Alborz mountains is represented by the grid cells 51 to 51.75E, 36.15 N. The results discussed are based on a 30-year climatological mean. For further details on this simulation the reader is referred to Jungandreas et al. [63, 65].

### 3.3. Modern observations

As modern reference climate, we use the Climate Research Unit observational time-series dataset of variations in climate, version 4.0 (CRU TS4.0, [66]), for the period 1961–1990. The data set has a resolution of 0.5˚x0.5˚. We choose the grid cells 51.25 to 51.75 E and 35.75 to 36.25 N as representative of the Jajrud catchment in the Alborz mountains and the grid cell with the center at 51.75 E and 35.25 N as the VP Fig 4.

### 3.4. Paleoclimate proxy data

Publicly available proxy data are scarce in the wider study region. The available archives and their proxies also differ regarding their natural setting as well as analytical parameters. Furthermore, the existing datasets have different temporal extents, time lags with respect to paleoclimatic conditions, and varying temporal resolutions and related uncertainties (specific age models), which makes the comparison between paleoclimate proxies very challenging [16, 67, 68]. Consequently, we decided to only use a very limited number of long-term and temporally well resolved speleothem records from Sofular [69], Jeita [70], Katalekhor [12], Tonnel'Naya [71], Talisman [72] and Mawmluh [73, 74] caves that are well distributed over the wider study area in order to compare them to the model outputs.

To work with a consistent dataset, we therefore use a new synthesis of pollen-based climate reconstructions, i.e. an extension of the LegacyClimate 1.0 dataset [75]. We use the version derived with the WA-PLS method [76]. The reconstructed annual mean precipitation (Pann) for 35 sites is interpolated on a common time axis with equally spaced 200-year time intervals using the basic R function "approx()" from the R package "stats" [77].

The paleoclimate records used in this study are mainly derived from lacustrine, peat, and alluvial/fluvial accumulations. The spatial representativeness of a record is mainly affected by the size of the pollen source area and thus by different sizes of lakes and peat areas. While paleoclimate records derived from lacustrine or peat archives represent a local (small lakes and peatland areas (< 1 ha); [78]) or a regional (larger lakes) scale, pollen assemblages from azonal riverine vegetation might be over-represented in fluvially impacted archives.

We can exclude a strong anthropogenic impact on the pollen records as abundances of Plantaginaceae and Rumex (indicators for grazing and animal husbandry) are low (< = 2% for

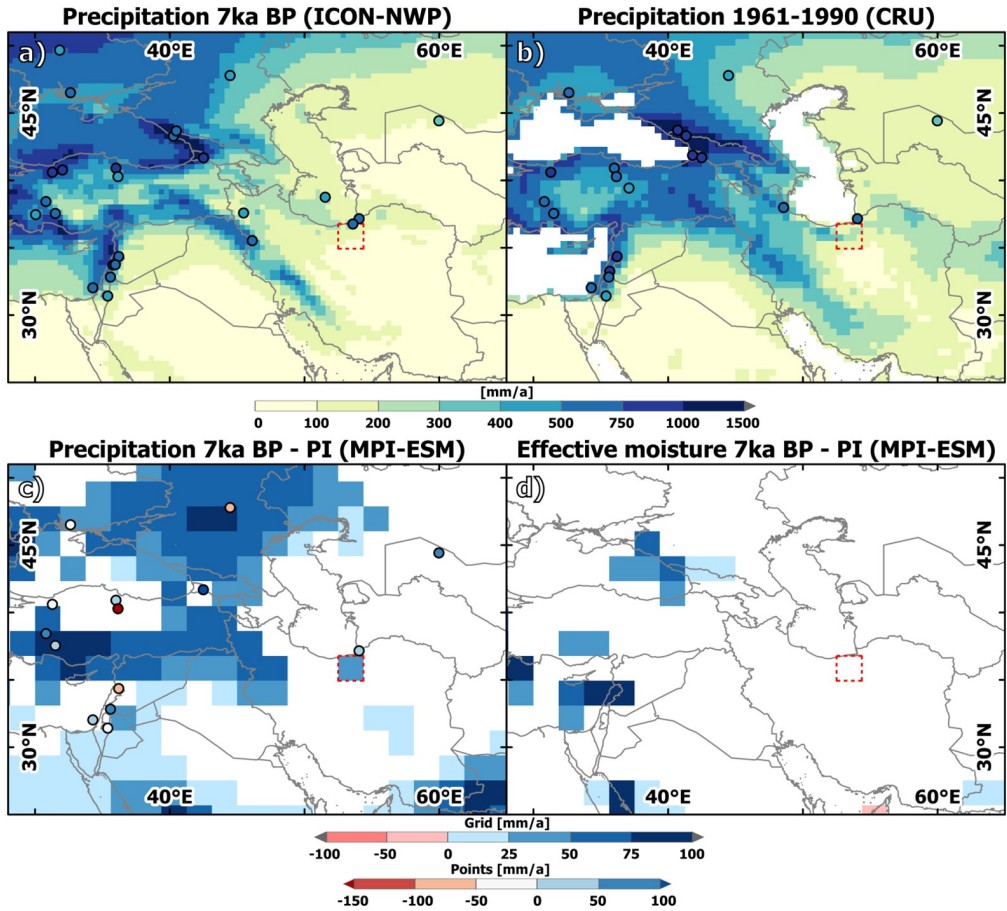

**Fig 4. Precipitation and effective moisture in Western Central Asia at 7 ka BP compared to today.** (a) Annual mean precipitation at 7 ka according to the ICON-NWP model (grid) and according to LegacyClimate2.0 (points, [85]). (b)observed annual mean precipitation over the period 1961–1990 based on CRU [66, 86] (grid) and according to LegacyClimate2.0 (points); (c) Significant (99%-level) differences in annual mean precipitation [mm/a] between 7ka and PI simulated by MPI-ESM (grid) and according to LegacyClimate2.0 (points); (d) Significant (99%-level) differences in effective moisture [mm/a] expressed as precipitation minus evaporation between 7 ka and PI simulated by MPI-ESM. The country and ocean boundaries are based on the feature layer "Global_Ocean_Country_Masks" (ID: 91) in ArcGIS Online. Note: The number of points in a), b) and c) varies according to the number of pollen records available in LegacyClimate2.0 for each respective time period.

*Rumex* and $< = 6\%$ for *Plantaginaceae*) for most sites used in this study for the Mid-Holocene (data not shown). Even in the Late Holocene (3 ka BP and younger), percentages of these taxa only rise up to a maximum of approximitely 18% at individual sites in the Eastern Mediterranean.

Like other reconstruction approaches, WA-PLS relies on extensive collections of modern surface samples. Both, the fossil and the modern pollen datasets are based on taxonomically harmonized assemblages, i.e. woody taxa were harmonized to genus level and herbaceous taxa were harmonized to family level. Although losing taxonomic information when merging taxa together into a higher taxonomic level, such an approach guarantees that all records are handled consistently.

Regression techniques like the WA-PLS method model relationships between pollen and climate. Those relationships are based on modelling assumptions such as the unimodality of the response of the pollen taxa to climate [76]. However, for limited taxonomic resolution by

merging several plant species with distinct climate requirements into one single pollen taxon or areas with an insufficient coverage of modern surface pollen samples, it might be difficult to create a calibration dataset that represents the required variety of environmental and climatic gradients. This may be indicated by high values in the root mean square error of prediction (RMSEP).

A proper reconstruction may struggle with intrinsic characteristics of the pollen compilations like a sensitivity to spatial autocorrelation or complex species responses, or in regions where fossil pollen are hardly preserved or nearby modern surface pollen samples are missing [79]. This is especially true for classification approaches like the Modern Analogue Technique (MAT; [80]) where limited analogues may produce poor results in so-called "quantification deserts" [81].

### 3.5. EOF analysis

As the different pollen records do not cover the same time periods, we calculate Recursively-Subtracted Empirical Orthogonal Functions (RSEOF) [82], using the R package "sinkr" [83]. The RSEOF method can additionally deal with datasets containing missing values [82]. All reconstructions with temporal extents of less than 1000 years that do not cover the Mid-Holocene or have an overly coarse resolution are excluded from this analysis. In order to make a model-data comparison that is as precise as possible, we prepared a pseudo-dataset based on the MPI-ESM model data that contains the same time steps included in the reconstructions and only the grid cells in which the pollen sites are located. Thus, we prepared a structurally identical dataset based on the model data to reflect the spatial-temporal variance of the reconstructions. In this study, only the first two EOFs are shown.

### 3.6. Time-slices and definition of reference periods

In addition to the analysis of the long-term trends in the transient simulation, we focus on different climatological time slices. Based on the available model outputs and the above-mentioned timing of archaeologically relevant periods of crisis, we define 7 ka BP as a starting or reference point for our study, corresponding to the Transitional Chalcolithic Period (cf. Tbl 1) associated with an increase in settlement density in most parts of the CIP. Furthermore, the time slice PI (pre-industrial, 1751–1850 CE) is defined as a reference for modern/current climatic conditions, also as a point of comparison to modern meteorological data. These periods are represented by climatological means over 100 years.

To shed light on the reasons for moisture extremes on the VP, we analyse a sequence of a wet and a dry periods in the model. To infer these periods, we calculate the moving mean over 100 years for different variables, i.e. annual mean precipitation, growing season soil moisture and the number of drought years (annual mean precipitation $< 150$ mm a$^{-1}$). The climate of these periods is then defined by the climatological mean of 50 years around the maximum or minimum extreme value of these variables, respectively. It is important to remember that the assignment of a drought to periods of reduced precipitation should only be done very carefully, since in modern hydrology, droughts are generally defined at an annual or at least less than decadal scale [84].

The significance of the simulated climatic changes between different periods is tested via a simple student's t-test. As significance level, $\alpha = .01$ is taken.

## 4. Results and discussion

The VP is located in what is currently a semi-arid region. Consequently, favourable living conditions are generally characterized by access to water resources. From a climatic point of view,

the water availability on the VP is governed by variations in precipitation either locally or upstream (Alborz Mountains) [50] or indirectly due to changes in temperature, which controls processes such as evaporation of water or the snowmelt in the mountains. We therefore concentrate on precipitation and temperature changes.

With regard to the model simulations, we divide the analysis of climatic changes into three distinct parts, i.e.

1. the analysis of the 7 ka time slice as representative of the Mid-Holocene climate in Western Central Asia when settlement density started to increase on the CIP;

2. the regional climatic variability in the period between 7 ka and 4 ka BP; and

3. the overall large-scale, spatio-temporal pattern of climate changes during the Holocene.

## 4.1. Paleoclimatic conditions around 7 ka BP—"The start/reference"

**4.1.1. Precipitation.**   During the Transitional Chalcolithic Period (approx. 7.3 to 6.3 ka BP), the VP (and also the CIP) experienced an increase in (rural) settlement density and sedentism [24]. As representative of this period, we analyse the 7 ka time slice that is not only covered by the MPI-ESM transient simulation, but also by the high-resolution ICON-NWP snapshot simulation. We compare the model results with pre-industrial and modern climate conditions.

According to MPI-ESM, WCA was rather wetter at 7 ka BP compared to PI Fig 4c, but the differences in the annual precipitation sum are mostly small (<50 mm) and not significant in large parts of the region. Substantially increased precipitation of up to 1000 mm a$^{-1}$ occurs only in the southeastern part (modern-day Pakistan and India) that is directly influenced by the intensified South Asian summer monsoon at the Mid-Holocene [42]. Little but significant precipitation enhancement at 7 ka BP compared to PI concentrates furthermore on the regions around the Black Sea and the Mediterranean Sea. In line with this pattern in rainfall change, the simulated 7 ka—PI differences in annual mean effective moisture, expressed in terms of precipitation minus evaporation, are only significant in parts of the South Asian monsoon region and the eastern coastal areas of the Black Sea and the Mediterranean Sea Fig 4d. According to MPI-ESM, the VP received only 30 mm a$^{-1}$ more rainfall at 7 ka BP compared to PI; the effective moisture does not differ significantly.

For the spatially higher resolved ICON-NWP simulation, no corresponding PI simulation exists. Therefore, we compare the output for 7 ka BP to a modern climatology, i.e. the period 1961–1990 in the CRU observational dataset [66, 86]. In contrast to MPI-ESM1.2, ICON-NWP shows rather dry conditions compared to the modern data at 7 ka BP in the eastern part of WCA, including the VP and the Alborz Mountains Fig 4a and 4b. Precipitation is enhanced and strongly concentrated on the western flank of the Zagros Mountains, presumably due to intensified orographic rainfall at 7 ka BP. In addition, the simulated annual mean climate is wetter at 7 ka BP compared to modern conditions around the Eastern Mediterranean Sea and north of the Black Sea. In line with the MPI-ESM simulation, the ICON-NWP model does not indicate substantial changes in the precipitation pattern in Western Central Asia for the 7 ka BP time slice. This result is also confirmed by the pollen-based precipitation reconstructions that show no clear pattern of increased or decreased annual mean precipitation at 7 ka BP. Only few reconstructions exist that cover the PI and 7 ka BP time slice, but at most of these sites, the 7 ka BP to PI changes ranges between ± 100 mm a$^{-1}$ Fig 4c and are therefore in a similar order of magnitude as the MPI-ESM simulation data. The pollen-based reconstructions reveal a relatively wet climate around the Caspian Sea at 7 ka BP, which may

point to the fact that the ICON-NWP model may underestimate the annual precipitation sum at the VP and in the Alborz Mountains. However, the reconstruction closest to the "Varamin grid cell" only reveals a slightly increased annual mean precipitation sum of less than 50 mm a⁻¹ at 7 ka BP compared to PI, agreeing well with the simulated change in the MPI-ESM model Fig 4c.

We conclude that the annual mean precipitation at 7 ka BP on the VP was on average similar to the present-day levels. However, precession-induced changes in the seasonal insolation led to a shift in the annual precipitation cycle Fig 5. This is revealed by outputs of both models used in this study. At present, the meteorological weather station at Tehran-Mehrabad Fig 3 and the CRU data for the VP Fig 5b reveal a rainy season from November to April, with maximum precipitation during winter. This seasonal cycle is well captured by MPI-ESM with the exception that the model strongly overestimates the November precipitation rate Fig 5a. For the 7 ka BP time slice, both models show a substantially increased spring precipitation, lasting from March to May in MPI-ESM and from April to June in ICON-NWP Fig 5. According to MPI-ESM1.2, the mean rainfall rate during the model month April is twice as high at 7 ka BP as in modern times (PI). ICON-NWP additionally simulates considerably lower rainfall during winter at 7 ka BP than in modern times, revealing a clear shift in the seasonal distribution. This shift may have been favourable for agricultural production in the VP as the increased local rainfall during spring at 7 ka BP would have resulted in increased levels of soil moisture during the main cultivation period. Consequently, according to MPI-ESM, the soil moisture is significantly increased from May to July in the VP at 7 ka BP compared to PI S3 Fig.

**4.1.2. Theories of different Mid-Holocene moisture regimes.** Different theories have been discussed so far as to why conditions on the VP and on the CIP at large were favourable for agriculture and settlement in the Mid-Holocene (e.g. [25]). In contrast, however, there are also quite a few climate model and proxy-based studies that suggest drier conditions than today (e.g. [36]) and therefore raise the question of why settlement density increased in this region during the early Mid-Holocene and whether this is really climate-driven. In the

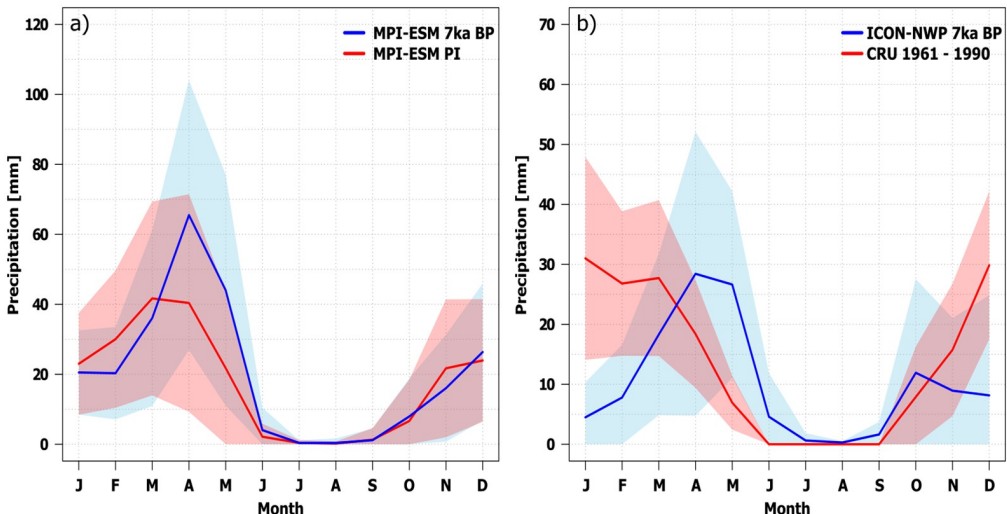

**Fig 5. Seasonal precipitation pattern for the grid-cells representing the Varamin Plain.** (a) Seasonal precipitation pattern at 7 ka BP and PI simulated by MPI-ESM [mm/month]. (b) seasonal precipitation pattern at 7 ka BP simulated by ICON-NWP in comparison to the modern observed seasonal cycle over the period 1961–1990 (CRU, [66, 86]). Note: +/- Standard deviation (shadings), based on a 100 yr sample (MPI-ESM) or 30 year sample (ICON-NWP and CRU).

following, we discuss some of these ideas and studies in the context of our model results. Specifically, we assess the role of a) the South Asian summer monsoon system, b) changes in the regional water supply by runoff and c) the Westerly wind system:

*4.1.2.1. South Asian monsoon influence.* Due to the change in seasonal insolation, the summer monsoon systems in the Northern Hemisphere were strongly enhanced and penetrated deeper into the continents during the Early and Mid-Holocene [42, 87, 88]. Particularly in the marginal regions of the monsoon domain, the intensified monsoon impact led to strongly increased precipitation. Could the VP or parts of the CIP have come under the direct influence of the monsoon circulation during early Mid-Holocene? The strongest precipitation change on the CIP between 7 ka BP and PI is simulated for late spring (April, May and partly June) S1 Fig, thus occurring during the pre-monsoon season. MPI-ESM simulates a substantially increased summer precipitation for large parts of the Arabian Peninsula (related to an extension of the West African monsoon), the margins of the South Asian monsoon and along the northern coast of the Indian Ocean. This is far away from our study area of the VP and the CIP. The model output of MPI-ESM, thus, does not indicate a direct influence of the summer monsoons on the rainfall in CIP during the early Mid-Holocene. To corroborate this result, a cluster analysis of seasonal precipitation in (modern day) Iran was performed (not shown), indicating no substantial change in the region dominated by summer precipitation over the course of the simulation.

*4.1.2.2. Intense runoff-feeding of the Jajrud alluvial fan.* The VP is located on the alluvial fan deposited by the Jajrud river that is fed by runoff from the Alborz and Anti-Alborz Mountains Fig 2. It is assumed that increased runoff during the Mid-Holocene could have favored living conditions on the VP [25]. Even nowadays, monthly mean temperatures are below 0˚C during winter in the headwater area, enabling snow accumulation on the top of the mountains (Fig 6). This snow melts during spring and increases runoff in the Jajrud river and, thus, in the VP. MPI-ESM reveals a generally cooler climate by up to 4˚C in most parts of WCA from November to May at 7 ka BP compared to PI (Fig 6 and S2 Fig). Following the orbital forcing, summer temperatures are higher throughout the region except for the area of the monsoon domain, where near surface temperature are lowered by extensive evaporative cooling. This enhancement of the seasonality is in line with findings by [89] for the Mid-Holocene (6 ka BP) in the Mediterranean area.

In the grid-cell mimicking the VP in MPI-ESM Sec 3.2.1, near-surface temperatures at 7 ka BP are approx. 2˚C lower during winter and spring than at PI and higher during summer, with a maximum difference in August of up to 4˚C. In combination with the shifted seasonality in precipitation (no significant winter rainfall changes but an enhanced spring precipitation), the milder climate during spring may have favored agriculture production by less soil moisture stress during the growing season. In addition, the substantially colder climate at 7 ka BP during winter may have led to more snow accumulation in the mountains and a later start of the snow melt, both enhancing late spring- and summertime runoff. We calculated the number of ice days in the ICON-NWP simulation, i.e. days with a mean daily temperature below 0˚C, for three grid cells in the Alborz Mountains representing the Jajrud catchment area Fig 6e. According to the model, a few ice days even occurred during April, in one particular year of the simulation a total of 8 ice days was simulated (in April). Furthermore, April is the month with the highest precipitation surplus at 7 ka BP. Both facts support the hypothesis of an extended season of snow accumulation and consequently more short-term storage of water in the Alborz mountains at 7 ka BP compared to modern times.

*4.1.2.3. Westerly wind regime shifts.* Numerous model- and proxy-based paleoclimate studies discuss the occurrence of drier conditions than today during the Mid-Holocene in WCA and also in arid Central Asia. The main argument is a shift of the Westerly winds including the

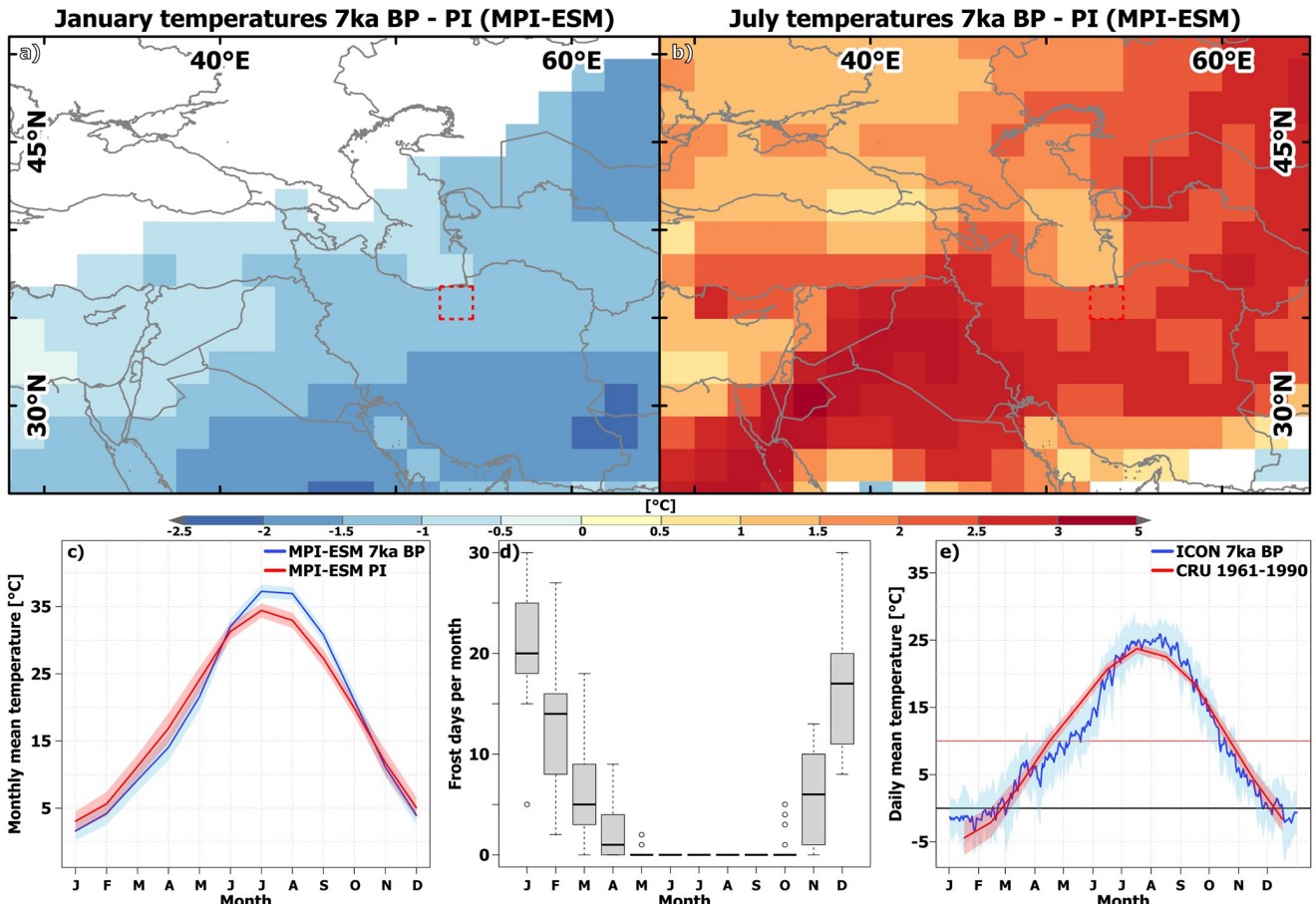

**Fig 6. Spatial and temporal comparison of temperatures in the study area between 7 ka BP and modern times.** (a) Simulated (MPI-ESM) differences in January temperatures between 7 ka BP and PI. (b) Simulated (MPI-ESM) differences in July temperatures between 7 ka BP and PI. (c) Seasonal temperature cycle in the grid-cell mimicking the Varamin plain in MPI-ESM at 7 ka BP and PI. (d) Number of frost days per month based on ICON-NWP simulation in the Alborz Mountains. (e) Seasonal temperature cycle on the Alborz Mountains (averaged over 3 and 4 grid-cells, respectively) according to ICON-NWP (7 ka BP) and in the modern observations (CRU, [66, 86]). The country and ocean boundaries are based on the feature layer "Global_Ocean_Country_Masks" (ID: 91) in ArcGIS Online. Note: Due to the different spatial resolution, the areas represented by ICON-NWP cells and CRU cells are not identical which causes differences in absolute values due to orographic effects on temperatures.

Subtropical Westerly Jet. In a high-resolution atmospheric model simulation series, Fallah et al. [36] found substantially drier winter climate in large parts of present-day Iran at 6 ka BP compared to PI, while the summer precipitation sum differs only in the monsoon-affected regions. Responsible for this is a shift in the simulated favourite jet stream position to approx. 54° N (north of Iran) at 6ka, leading to a weakening of the cyclonic activity and therewith to less precipitation. Over the course of the Holocene, the jet position moves southward coinciding with an increase in wintertime precipitation in the simulations by Fallah et al. [36]. Similar results have also been reached in a model-intercomparison study in the Palaeoclimate Modeling Intercomparison Project (PMIP) by Wang et al. [38]. According to the PMIP4 model ensemble, winter and spring precipitation was reduced over large parts of arid Central Asia, including WCA, comprising the CIP and the VP, at 6 ka BP. Due to an insolation-induced decrease in the winter and springtime meridional temperature gradient at 6 ka BP, the Westerly winds and the moisture transport in the Westerly winds from the Mediterranean Sea and the North Atlantic are weaker in the models. In addition, the cooler climate limits the local

moisture recycling and evaporation from upstream water bodies. Thus, less moisture is transported to Central Asia by the Westerlies.

Paleoclimate reconstructions based on different kinds of archives partly support the modelling results and reveal drier Mid-Holocene conditions compared to PI and a persistent increase in moisture towards the present in the core region of arid Central Asia [38, 90]. Other interpretations of the few records propose a Mid-Holocene moisture optimum in arid Central Asia [91].

While these syntheses provide a relatively clear picture of climatic changes in eastern Central Asia, reconstructions for Western Central Asia are rare. Based on pollen and oxygen-isotope records, Jones et al. [30] deduce the wettest period during the Holocene for the Zagros Mountains and northwestern Iran from approx. 7 ka to 5 ka BP. Stevens et al. [29, 40] interpret the changes in the oxygen-isotope records at Lake Zeribar and Mirabad located on the northwestern Zagros mountain chain to an overall enhanced precipitation and a shifted seasonality from wintertime to springtime precipitation, which is in line with enhanced precipitation in MPI-ESM during spring at 7 ka BP. For the Eastern Mediterranean region, studies based on diverse paleoclimatic proxies reveal a wetter Mid-Holocene compared to the present, that has been linked to shifts in the Westerly wind regime [92].

The studies presented above indicate that the Central Asian region, including the WCA region, is climatically complex and that the precipitation signal is highly sensitive to the position of the Westerly jet tracks. The CIP seems to be in a transition zone of a rather moist Mid-Holocene climate to the west and a drier climate to the east compared to present conditions. The annual average signal seems to be determined mainly by the balance of increased spring- and decreased wintertime precipitation. A comparison of the PMIP3 simulations reveal wetter springs at 6 ka BP than at PI south of approx. 38° N in WCA, and drier conditions north of it [93], thus indicating that the MPI-ESM results retrieved in this study are not contradictory to the PMIP simulations.

However, MPI-ESM reveals no substantial change in the position of the Westerly Subtropical Jet, nor of the Westerly wind band position over WCA (Fig 7). The upper-tropospheric Westerly winds (250 hPa) are weaker during the months February to April at 7 ka BP over large parts of WCA including the CIP, while they are stronger on the southern part of the Arabian Peninsula and the northern Indian Ocean. The diminished Westerly wind speed at 7 ka BP in June south of 32° N and in the entire WCA during August and September reflects the stronger Tropical Easterly Jet compared to PI. During October and November, the Westerly wind strength is increased north of 30° N and decreased south of 30° N at 7 ka BP compared to PI.

The inverse pattern with respect to the spring (enhanced at 7 ka BP) and November (decreased at 7 ka BP) precipitation anomalies, reveals no direct link of the Westerly wind speed anomaly to the precipitation change on the CIP and the VP. It is difficult to determine exactly what mechanisms led to increased precipitation in 7 ka BP in MPI-ESM because it is a dynamic system including internal feedback mechanisms. The atmospheric flow in the upper troposphere (250 hPa) shows a cyclonic (counterclockwise), mostly divergent wind anomaly with its core around the Persian Gulf (not shown). This wind anomaly is accompanied by enhanced rising air masses in the middle troposphere which favour precipitation formation. On the contrary, during November, the upper tropospheric wind field is characterized by an anticyclonic (clockwise) wind anomaly with enhanced subsidence in the middle troposphere, suppressing precipitation (not shown).

**4.1.3. Synthesis.** Compared to present day climatic conditions, the model results do not indicate specifically favourable climatic conditions at 7 ka BP that would be able to satisfactorily explain the increased settlement density on the VP during early Mid-Holocene. Mean

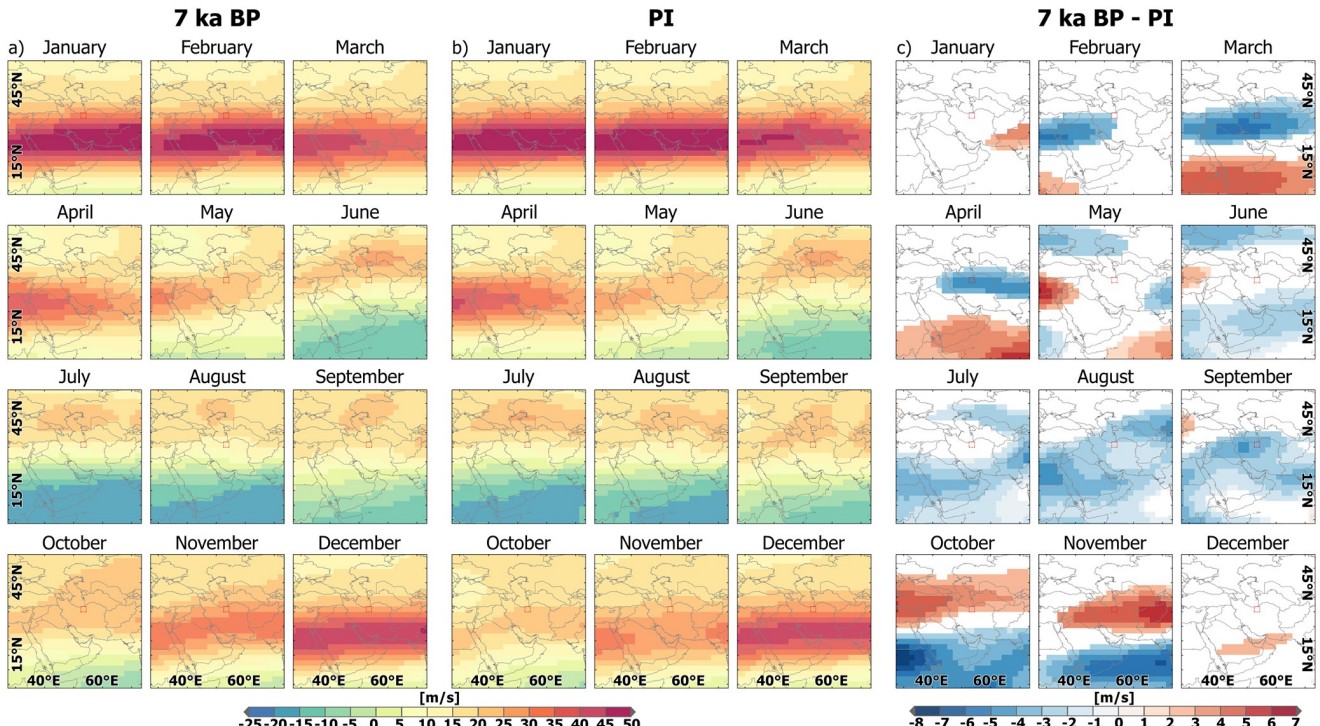

**Fig 7. Zonal wind component in Western Central Asia simulated by MPI-ESM.** (a) Zonal wind component at 250 hPa at 7 ka BP simulated by MPI-ESM. (b) Zonal wind component at 250 hPa at PI simulated by MPI-ESM. (c) and significant changes between in the zonal wind component at 250 hPa 7 ka BP and PI simulated by MPI-ESM. The country and ocean boundaries are based on the feature layer "Global_Ocean_Country_Masks" (ID: 91) in ArcGIS Online. Note: red colours—Westerly wind directions, blue colours—Easterly wind directions.

annual precipitation and moisture levels are similar to the present. In line with previous paleo-climatic proxy data and model derived conclusions, MPI-ESM and ICON-NWP reveal a changed seasonality at 7 ka BP. Higher mean precipitation rates during the vegetation period in spring in combination with colder and longer-lasting winters fostering water storage in the Alborz and Zagros mountains may have favoured water supply and availability for agricultural production. However, it is unlikely that these comparatively small absolute changes would have fundamentally contributed to a reliable and continuous source of water for livelihood and agriculture. They could, however, have been one of the many reasons for the spread of rural settlements in the Transitional and especially the Early/Middle Chalcolithic periods. Most likely, people relied to a large extent on springs and perennial river channels associated with alluvial fans [23].

## 4.2. Climate variability on the varamin plain during the mid-holocene

Changes in settlement patterns are often discussed as being related to the impact of strong climatic events such as intense drought periods [3, 9, 14, 17, 28]. Shaikh Baikloo Islam & Amir-khiz [5] identified several "climatic events" on the CIP during the Mid-Holocene which they associate with changes in settlement patterns, among them two drought periods at approx. 5.2 ka and 4.2 ka BP.

While in some areas of Iran (e.g., the central Zagros and western parts of Iran), the 5.2 ka BP event is reflected in paleoclimatic proxy records and correlates with the local decrease in settlements [94, 95], the climate variability on the CIP and northeastern plateau is not in line

with the settlement dynamic during this period [17]. The same disparity between the developments in climate and settlement density is observable for the 4.2 ka event. Although identified in many records worldwide, the 4.2 ka event is not visible in all regional climate reconstructions in WCA. The central and western parts of Iran faced a decrease in settlements, but the decline often started earlier and no synchronous collapse of cultures occurred at 4.2 ka BP. In some parts, the settlement density even increased around this period [17].

As a consequence of these partly overlapping and partly contradictory records and trends, paleoclimatic interpretations overall remain vague for the region, and there is no evidence of a clear regional homogenous correlation of settlement dynamics and climate variability. Climatic events seem to affect settlement dynamics rather locally. Other regions seem not to be susceptible to droughts. This regionally diverse picture may be partly related to a spatially heterogeneous response in climate to external or internal climate forcing and high internal climate variability due to, e.g., the local environmental conditions.

From a methodological point of view, the comparison of settlement datasets with climate data (almost exclusively derived from proxies) has been carried out on different spatial (and temporal) scales for the Mid- and Late Holocene. While in some recent studies, rather large areas (differentiated into sub-areas) have been analysed [96], other studies focus on sub-regions [97, 98]. In some recent studies, results of archaeological surveys are combined with summed probability distributions (SPDs) of radiocarbon dates as proxies for settlement densities. Some of the challenges related to the spatial scale, the constraints caused by issues of chronology and other methodological issues such as limited research budgets, visibility of diagnostic artifacts and variations in statistical methods have been discussed elsewhere [9, 96]. We consider our study as a pilot study, based on the comparatively low spatial resolution of the climate dataset (despite its comparatively high resolution given the temporal extent and resolution, see Sec 3.2) but with the advantage of being a global atmospheric model, compared to the comparatively small survey area on the VP. We have tried to overcome these challenges by applying three different spatial scales and verifying the model results with proxy reconstructions. While this approach will not able to reconcile heterogeneous or even contradictory archaeological or paleoclimatic proxy datasets, it may contribute to the differentiation between rather large-scale (temporal and spatial) atmospheric causes or changes in climate and more local or regional conditions and causes for changes in settlement dynamics.

**4.2.1. Simulated dry and wet spells.** To explore the climatic variability on the VP, we analyse the simulated (MPI-ESM) changes in different indicators that could cause drought or moisture stress, i.e. the local precipitation change, the change in soil moisture during the growing season and the change in the number of particularly dry periods. For this study, we define the latter as the climatological mean over 100 years with annual precipitation sums below 150 mm. A precipitation sum of 150–200 mm a$^{-1}$ is seen as the minimum limit for natural steppe development [99].

The local climate reveals a high degree of temporal variability. The simulated annual mean precipitation sum on the VP ranges from about 50 mm a$^{-1}$ to 500 mm a$^{-1}$, but climatological changes are small over the course of the Holocene Fig 8a. Around 7.2 ka BP, the model simulates a pronounced dry period, seen in all indicators, followed by a period with more humid conditions with maximum climatological mean precipitation and high soil moisture around 6.4 ka BP. This is slightly later than the climate optimum recorded for the broader region around 5500–5000 BCE (approx. 7.5–7 ka BP) based on paleoclimatic reconstructions [30], but it still coincides with the increase in settlement density on the VP. Subsequently, a period with a generally more arid climate and less pronounced variability in precipitation is simulated. However, the soil moisture and drought years show a larger imprint of the wet and dry phases as in the period before. The period around 4.5 ka to 4 ka BP is relatively wet in the

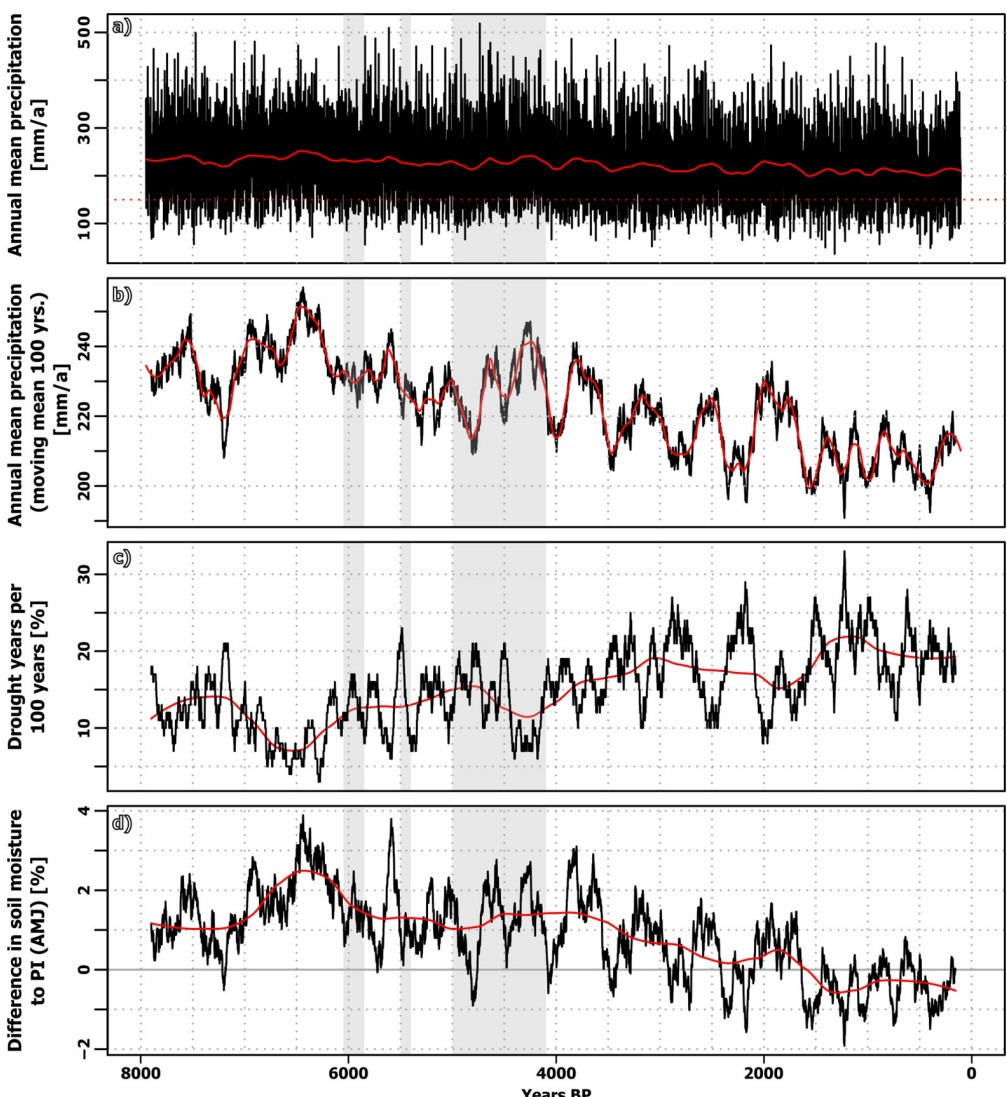

**Fig 8. Time-series of simulated precipitation, drought years and soil moisture on the Varamin Plain (based on MPI ESM).** (a) Simulated annual mean precipitation (annual resolution, black line) + smoothed by LOESS filter (red line). (b) Annual mean precipitation as 100-year moving mean (black line) + smoothed by LOESS filter (red line). (c) Moving mean of the number of drought years (= precip < 150mm/a) within 100 years. (d) Difference in soil moisture [%] for the months April-June, compared to PI as moving mean (black line) and smoothed by LOESS filter (red line). Note: All time-series are displayed for the grid-cell representing the Varamin Plain; grey areas represent periods of crisis on the Varamin Plain. For monthly information, the modern calendar is used.

model data and therefore not in line with most paleoclimatic proxies (although there are substantially contradicting results for this period in the paleoclimatic proxy data) as well as with low settlement density in most parts of the CIP [17]. The Late Holocene is characterized by a strong variability with pronounced and long-lasting wet and dry spells roughly every 500 years, overlapping a general drying trend seen in all analysed moisture indicators. Particularly, the number of drought years per climatological period substantially increases Fig 8.

We choose the simulated wet period around 5.6 ka BP and the simulated dry period around 5.5 ka BP as examples of possible climatic events and analysed the circulation changes between both as they roughly coincide with the timing of a (settlement) crisis on the VP. Both phases

are reflected in all drought indicators considered here. The two periods differ significantly, although the overall/absolute differences are rather small. Most of the changes in annual mean precipitation are related to a decreased precipitation rate in the model month April (not shown).

During April in the dry period at 5.5 ka BP, the air mass above the Tibetan Plateau is substantially colder than in the wet phase, inducing a cyclonic (counter-clockwise) atmospheric circulation anomaly in the upper troposphere (250 hPa) around the Tibetan Plateau Fig 9. In

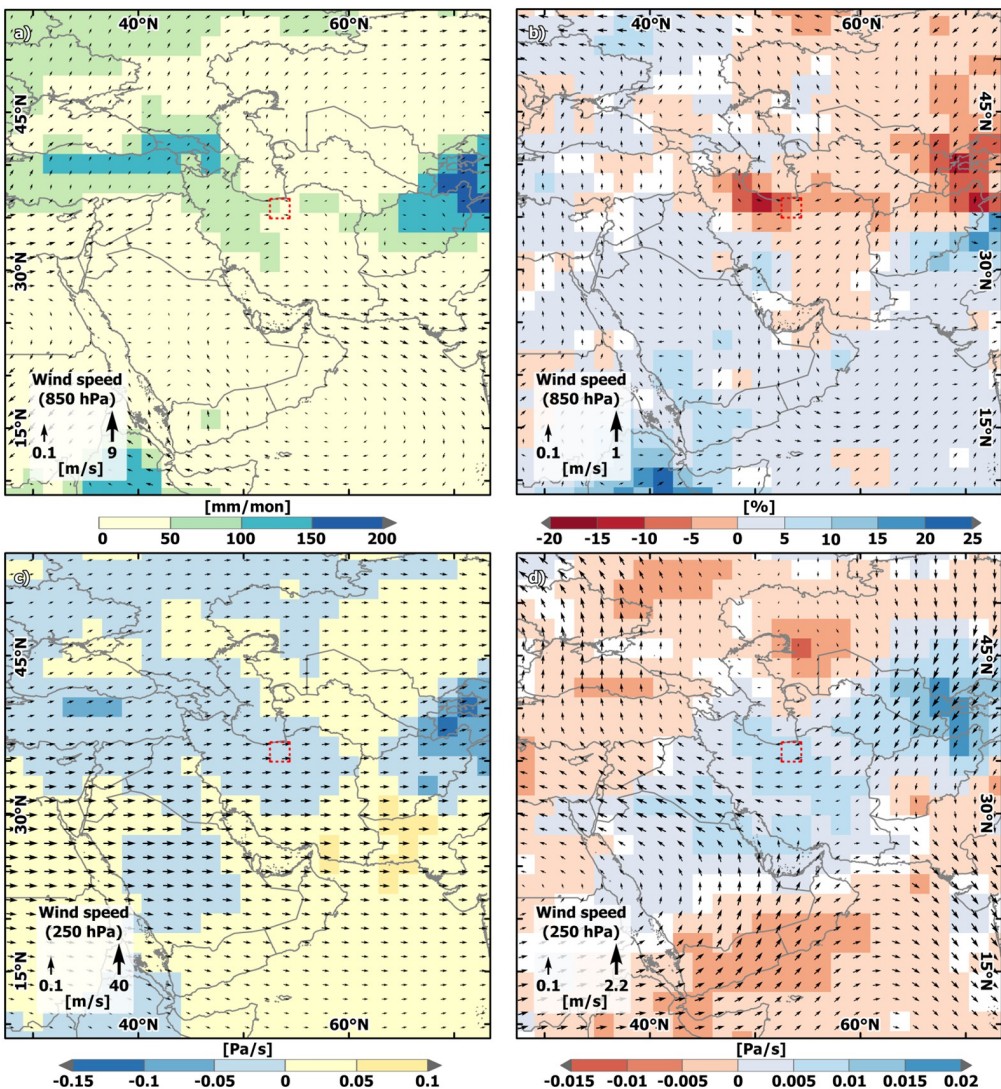

**Fig 9. Simulated precipitation and wind fields in Western Central Asia during Mid-Holocene wet and dry periods.** (a). Simulated precipitation [mm/month] (shaded) and low-atmospheric wind field (850 hPa, vector) [m/s] for April averaged over a dry period (~ at 5.5 ka BP, see text for details). (b) Percent differences between a dry period (~at 5.5 ka BP, see text for details) and a wet period (~5.6 ka BP) in simulated precipitation [mm/month] (shaded), and the difference in low-atmospheric wind field (850 hPa, vector) [m/s] for April. (c) Mid-troposheric vertical motion at 500hPa (yellow and red = descend, blue = uplift, shaded) [Pa/100 s] and upper-tropospheric wind field (250 hPa, vector) [m/s] for April during a dry period (~ at 5.5 ka BP, see text for details). (d) Differences between a dry period (~at 5.5 ka BP, see text for details) and a wet period (~5.6 ka BP) in simulated mid-tropospheric vertical motion at 500hPa (shaded) [Pa/100s] and the upper-tropospheric wind field (250 hPa, vector) [m/s] for April. The country and ocean boundaries are based on the feature layer "Global_Ocean_Country_Masks" (ID: 91) in ArcGIS Online.

contrast, the air above the Caspian Sea is warmer during the dry event, causing an anticyclonic (clockwise) anomaly in the upper tropospheric (250 hPa-level) circulation. Both induce strong north-easterly wind anomalies east of the Caspian Sea and easterly wind anomalies above the Zagros mountains, substantially reducing the intensity of the upper-tropospheric (and also low-level) Westerlies above this region, including the CIP and VP. The position of the Jet stream and the Jet streak do not differ between the wet and the dry phases. This reduction in intensity of the Westerlies coincides with a subsidence anomaly. During April, the CIP is located below the left exit of the Jet Streak which is known to favour ageostrophic circulations with wind divergence in the upper level and an uplift of air-masses below, cyclogenesis and enhancement of precipitation. This ascent is also visible in the simulated dry phase at 5.5. ka BP, but substantially reduced compared to the wet phase. Along the eastern flank of the Zagros mountains, this leads to a decreased monthly mean precipitation by 10–30% compared to the wet phase. In the grid cell mimicking the VP, rainfall is reduced by 20%. In the north-western part of the Zagros and also the Eastern Mediterranean Sea domain, precipitation does not change. Over the northern Indian Ocean and the Himalayas, the highly divergent upper tropospheric wind field in the dry period coincides with a vertical upward motion anomaly and an increased precipitation in the dry phase. The low-level atmospheric flow anomaly over the Indian Ocean resembles a monsoon-like circulation, but no indicator exists for an earlier onset of the monsoon season in the model, nor that the monsoon system is the driver of the atmospheric wind anomalies in WCA during the simulated dry period at 5.5 ka BP. The contrasting precipitation response of the monsoon area, the eastern Mediterranean and the eastern part of the WCA point to the fact that drought events must not have a spatially homogenous footprint. In particular, the Zagros mountains seem to act as a separator for the different responses.

**4.2.2. Synthesis.** The model data shows several dry and wet spells during the Holocene in the VP and a high climatic variability in this region, even though the overall climatic change during the Holocene is small. Events like the proposed 5.2 ka event or the 4.2 ka event are not simulated in the model. The drivers of these events are still debated and the subject of ongoing research. Possible causes for the 4.2 ka climate anomaly, such as Bond events in the North Atlantic, are not prescribed in the model as forcings. In addition, the model has its own internal climate variability. Therefore, we do not expect reconstructed climatic events to be reproduced by the model exactly at the times in which they appear in the proxy-based paleoclimatic records. However, the model demonstrates that prolonged periods of drought may have existed during the Holocene, which could then in principle also have affected settlement dynamics. These drought periods are probably related to particularly strong reductions in the Westerly winds, but do not cover all of WCA. In the example given here, temperature anomalies above the Caspian Sea and the Tibetan plateau are the main drivers. The latter points to the fact that processes on the Tibetan plateau, such as changes in snow cover, may substantially alter the upstream Westerly winds and therewith the climate in WCA.

## 4.3. Spatio-temporal precipitation change during mid- and late holocene

Proxy- and model-based deductions reveal a complex response of the climate system to the Holocene insolation forcing in Central Asia. While some studies point to a steady increase in humidity towards the present, e.g., in parts of Central Asia [90], in other studies, e.g. for the eastern Mediterranean and regions directly influenced by the monsoon, a general drying trend is attested for the transition from the comparatively humid Early Holocene to the Mid- and Late Holocene [16]. This millennial-scale trend is attributed to shifts in the intensity and the location of the Westerly Jet, cyclonic activity and monsoonal rains (as discussed above).

In order to set the precipitation changes in the VP into the context of the spatio-temporal pattern of precipitation changes, we performed a Recursively-Subtracted Empirical Orthogonal Function analysis (RSEOF). For this, we used the MPI-ESM transient simulation and the pollen-based LegacyClimate 2.0 dataset and extended the WCA region further to the west, including the reconstruction upstream of the Westerly wind band. Both model and pollen-based reconstructions enable a consistent analysis of the spatio-temporal precipitation change in the region. We compared the first two principal components (PC) with isotope data (δ18O) that are often used as an indicator of changes in precipitation [12, 16, 100] and often possess a relatively high temporal resolution.

**4.3.1. Principal component 1/EOF 1.**    The first PC explains 43% of the total variance in the LegacyClimate2.0 dataset and indicates fairly stable and comparatively humid conditions from 8 to 4 ka BP. This is followed by a sharp aridification trend until 1.5 ka BP and another short and more humid phase in the last millennium BP. The palynological records along the southern Caspian Sea which are closest to the VP as well as records from the Levant reveal positive loadings for this PC, while palynological records from the Black Sea, modern-day Turkey and the Zagros mountains show negative loadings, thus the opposite trend Fig 10.

The first PC in the model data explains 70% of the total variance, which is a much higher value as the 43% explained by the first PC of the pollen-based reconstruction. With respect to the temporal trend, it resembles the first PC of the pollen data, with wet conditions during the early Mid-Holocene and a drying trend towards PI. However, the decrease in precipitation already starts at 7 ka BP in the model and is much smoother than in the pollen-based reconstructions. This may partly be related to the fact that the vegetation response may lag behind the climatic response by several thousand years, as discussed for pollen records on the Zagros mountains ([16] and references therein). However, it is currently unclear to what extent a lag in vegetation (foremost trees) is reflected in pollen-based climate reconstructions (c.f. [101]).

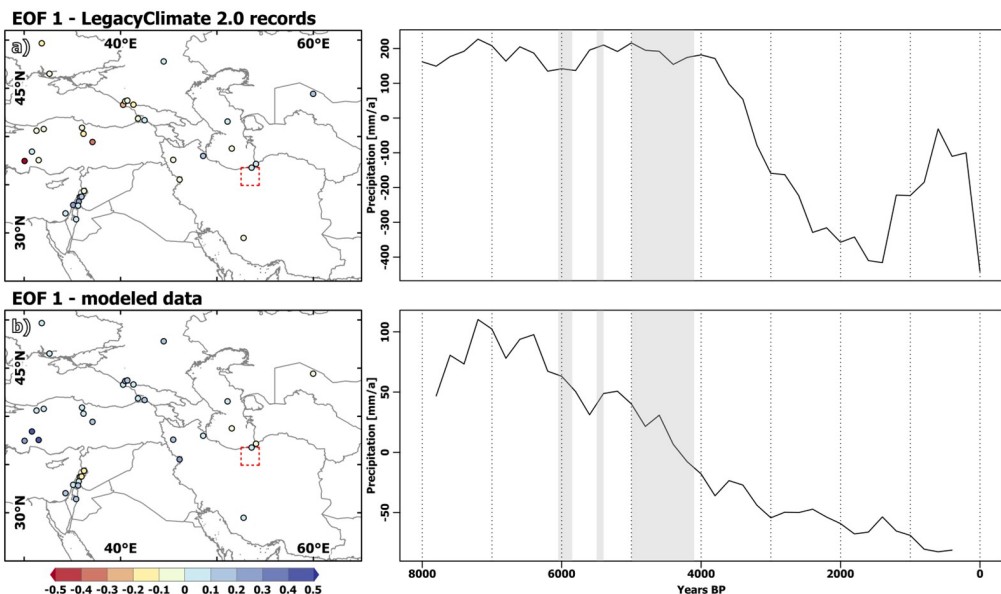

**Fig 10. First Empirical Orthogonal Function of pollen derived climatic reconstructions and model data.** EOF analysis, 1st Empirical orthogonal function (EOF) with colour coding indicating loadings of EOF and 1st principal component (PC1) (time-series) for the annual mean precipitation signal (a) in the LegacyClimate 2.0 records and (b) model derived "pseudo"-records. The country and ocean boundaries are based on the feature layer "Global_Ocean_Country_Masks" (ID: 91) in ArcGIS Online.

In contrast to the pollen data, the model data reveal positive loadings nearly everywhere in the extended WCA region, reflecting a spatially more consistent and homogeneous response in the model. This may be driven at least partly by the much smoother orography in the model, in which local environmental changes recorded in the pollen samples cannot be captured.

Reconstructions of the δ18O isotopes based on the few available long-term speleothem records in the broader region indicate higher negative values during the Mid-Holocene compared to the Late-Holocene Fig 11 that can be interpreted as wetter conditions during the Mid-Holocene, although it is highly debated which signal is recorded in this kind of proxy [16]. The co-occurring increase in δ18O in all records towards PI generally confirms the results of the model with respect to a more homogenous response. In particular, the nearest available speleothem record from Katalekhor Cave in Iran [12] agrees well with the model-derived first PC, showing an increase in moisture level between 7.5 ka and 7 ka BP and a decrease in precipitation afterwards.

**4.3.2. Principal component 2/EOF2.** The second PC explains 20% of the variance in the LegacyClimate2.0 precipitation reconstructions Fig 12a. In contrast to the first PC, the second PC indicates rather dry conditions until about 5.5 ka BP, followed by a marked increase in humidity until 4 ka BP. After a fairly continuous humid phase until about 2 ka BP, the second PC reveals rather dry conditions for the last two millennia. No clear spatial distribution with respect to the loadings of this PC is apparent. The highest loadings are in fact represented by the records along the southern shore of the Caspian Sea, while records from the eastern Mediterranean and the regions around the Black Sea show contradicting loadings on a regional scale. These findings confirm results by Rafiei-Alavi et al. [17] with respect to regional heterogeneity in the proxy records. Interestingly, the second PC derived from the pollen-based reconstructions reveals the regionally optimal conditions during the late Mid-Holocene and early Late Holocene for the Gorgan Plain (southern eastern margin of Caspian Sea). It has been proposed that settlement density on the CIP may have decreased from approx. 5 ka BP onwards, because people migrated towards the Gorgan Plain via the lower heights of the Alborz Mountains [102]. The Gorgan Plain experienced an increase in social complexity and in the number of settlements at the beginning of the Bronze Age [103].

However, in a multi-proxy-study by [104] at Kongor Lake in the eastern Gorgan Plain, a short comparatively wet period between 6.1 and 5.9 ka BP fades into more arid conditions including increased fire frequency/intensity and desiccation events between 5.9 and 3.9 ka BP, which is in line with different studies attesting to a sinking sea level in the Caspian Sea during this time [105]. In contrast, in some records [106, 107], short wet(ter) periods around 5 ka BP have been identified, adding to the controversy about climatic conditions during this period.

The second PC for the model data Fig 12b shows a similar temporal change, although the trend towards more humid conditions starts already around 6.5 ka BP and proceeds much more rapidly than in the pollen-based reconstructions. The second PC explains only 13% of the variance in the simulated precipitation signal. Loadings are most positive in the region around the eastern Black Sea, while the loadings at the eastern Mediterranean coast are mostly negative. Diverging loadings are revealed for the Caspian Sea, indicating–in line with the reconstructions–a regionally heterogeneous response.

**4.3.3. Monsoon vs. westerly response.** The overall long-term change represented in the first and second PCs resembles the simplified moisture evolution pattern derived by [92] and [90] for South and Central Asia. The first PC co-evolves with the response of the South Asian summer monsoon to the Holocene orbital forcing, reflecting a decrease in monsoon intensity and related precipitation from about 8 ka BP until present. The second PC is roughly in line with the signal in arid Central Asia, presumably reflecting a regime shift in the upper level

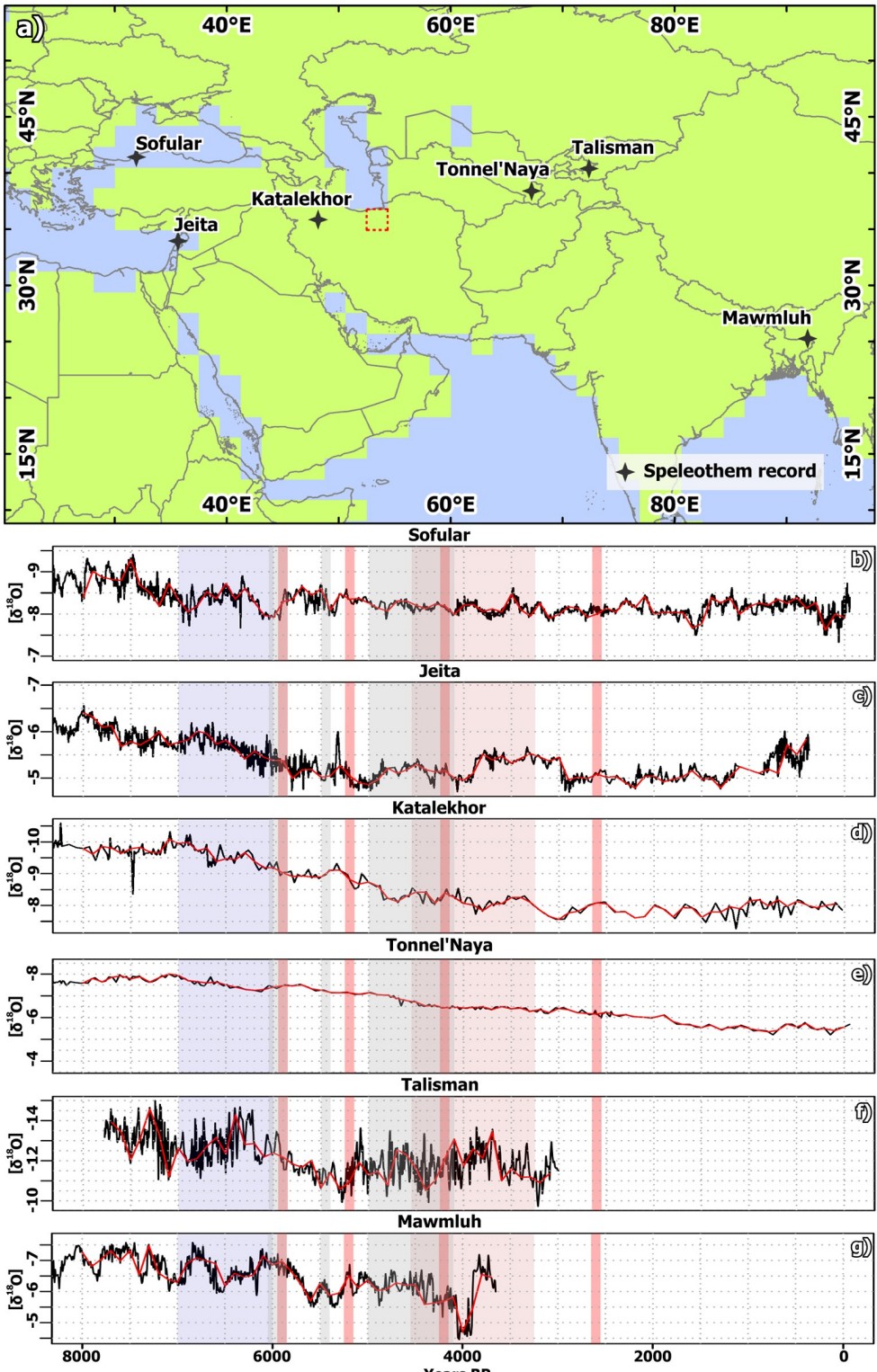

**Fig 11. Location and δ18O time-series of six long-term speleothem records in Western Asia.** (a) Location of six long-term speleothem records in Western Asia before the background of land-sea-mask in MPI-ESM and modern national borders. (b-g) δ18O curves of speleothems in WCA (for references to original data see section Sec 3.4; grey areas represent periods of crisis on the Varamin Plain. The blue areas represent a particular wet period reported in literature and the red areas indicate dry events and phases reported in literature for the region (seeSec 4.2). The

country and ocean boundaries are based on the feature layer "Global_Ocean_Country_Masks" (ID: 91) in ArcGIS Online. Note: δ18O-values are negative values in order to achieve visual correspondence to precipitation curves (as more negative δ18O-values are associated with wetter conditions and therefore higher precipitation amounts).

Westerly wind Jet stream. The EOF analysis reveals that both responses are part of the regional precipitation change during the Holocene, albeit less pronounced in the model data. Which mode dominates at certain sites may be controlled by the local environmental conditions such as topography, that substantially affect the regional Westerly wind dynamics and the precipitation in WCA [51] and the eastern Mediterranean. This includes the teleconnection response to the strong diabatic heating in the South Asian monsoon domain, known as the Rodwell-Hoskins Mechanism [108]. The strong monsoonal heating induces Rossby-waves that cause subsidence west of the monsoons (e.g., eastern Mediterranean) with cyclonic circulation in lower levels and anticyclonic circulation in upper levels of the troposphere. The location of this descent is partly determined by the Zagros mountains [109] that reinforce the subsidence over the central and eastern Mediterranean and in the region south of the Aral Sea (Turkmenistan/Uzbekistan). In contrast, the mountains foster ascent above their location. The Zagros Mountains have the strongest impact when low-level atmospheric flow is easterly, i.e. during the summer [109]. The Holocene changes in summer monsoon intensity thus affect the humidity in the subtropical regions by modifying the vertical motion and therewith the preconditions for the formation of precipitation. In addition, changes in the seasonality of the Westerly wind system (by prolonging easterly wind conditions) or of the monsoon system (by modifying the onset or withdrawal of the monsoon) can have a decisive influence on the precipitation during other seasons. Differences in orography and rainfall seasonality may thus at

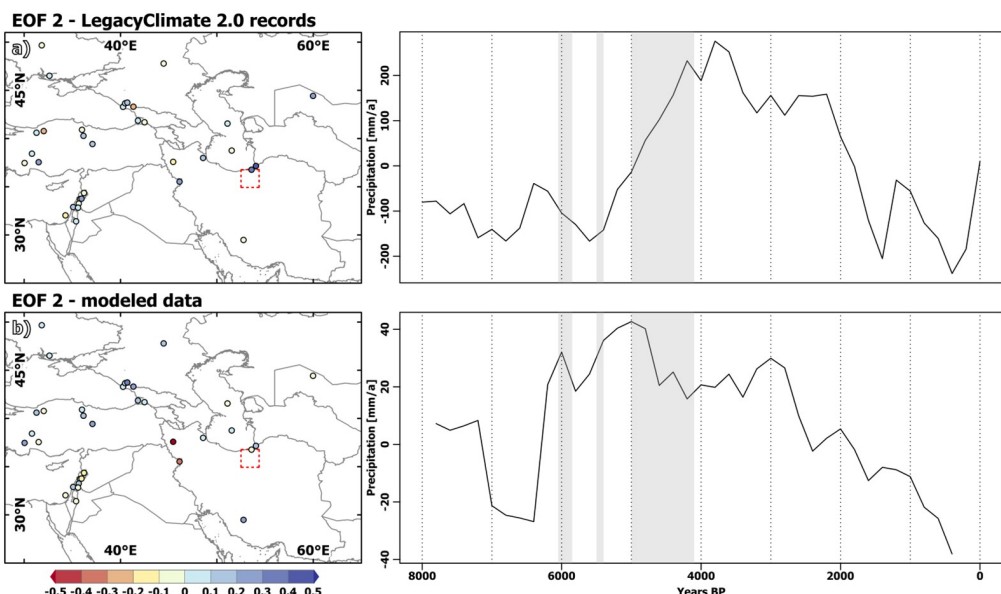

**Fig 12. Second Empirical Orthogonal Function of pollen derived climatic reconstructions and model data.** EOF analysis, 2nd Empirical orthogonal function (EOF) with colour coding indicating loadings of EOF and 2nd principal component (PC1) (time-series) for the annual mean precipitation signal (a) in the LegacyClimate 2.0 records and (b) model derived "pseudo"-records. The country and ocean boundaries are based on the feature layer "Global_Ocean_Country_Masks" (ID: 91) in ArcGIS Online.

least partly be responsible for the spatial differences in the EOF loadings. In some individual cases, local anthropogenic impacts, especially for the Late Holocene, cannot be excluded.

Since the model simulates no shifts in the Westerly jet position but rather in the strength of the upper level Westerlies, the monsoon-mode is much more present in the model than in the pollen data, explaining 70% of the variance. This is in line with previous findings of a dominant imprint of Holocene changes in global monsoon intensity on global precipitation change (cf. [42]). To infer which mechanisms lead to the in-phase response of the monsoon system and the WCA region in the model would require a set of sensitivity experiments that goes beyond this paper. Since the precipitation changes in the VP occur mainly during spring, we imagine variations in the seasonality of the monsoons and related teleconnections as possible drivers.

## 5. Conclusion and outlook

The Varamin Plain (VP), an alluvial fan east of Tehran (Iran), experienced an increase in settlement density beginning at approx. 7.3 to 6.3 ka BP, faced several settlement discontinuities in the following millennia and was finally abandoned for several centuries (approx. 4.9 to 4.1 ka BP). Settlement crises are often assumed to be associated with (abrupt) climate events such as droughts. However, the number of continuous proxy records in WCA is low and the existing ones reveal partly contradictory and non-uniform changes over time, leaving the climatic interpretation and possible relationships to settlement dynamics unclear.

In this study, we therefore explore the regional climatic development from different perspectives, using a transient Earth System Model simulation performed in MPI-ESM [42], a high-resolution snapshot simulation by ICON-NWP [110] for the Mid-Holocene and pollen-based climate reconstructions from LegacyClimate 2.0 [85].

Overall, the modelled precipitation sum and moisture availability averaged over the year are quite similar to modern conditions, thus, not revealing particularly favourable climatic conditions that may satisfactorily explain the increased settlement density on the VP during the early Mid-Holocene. However, the model MPI-ESM indicates the wettest conditions during the last 8000 years for this time slice. Additionally, both models (MPI-ESM, ICON NWP) reveal substantial shifts in the seasonal rainfall and temperature cycle. Colder winters and springs and increased spring precipitation enhance the soil moisture during the vegetation period and foster snow accumulation during the cold season and a delayed snow melt on the Alborz Mountains. A resulting increase in runoff during the summer could have led to a greater water supply on the alluvial fan(s) during the vegetation period, additionally favouring agricultural production.

The MPI-ESM model reveals a high year-to-year variability (50–500 mm $a^{-1}$) and simulates a sequence of dry and wet periods for the VP. These periods are driven by changes in the intensity of the Westerly winds with substantially decreased upper tropospheric Westerly winds and subsidence over the eastern part of the WCA during dry periods in the model, impeding the formation of precipitation. The model indicates that drought events perceived in the VP have no spatially uniform manifestation. Rather, the Zagros Mountains seem to represent a spatial boundary that evokes contrasting responses between the region to the east and the region to the west of the mountains. The model results furthermore reveal that changes on the Tibetan Plateau can substantially impact the circulation upstream in WCA. This is particularly relevant with respect to future climate change on the Tibetan Plateau that may, e.g., lead to a substantial reduction of the snow cover in this highly sensitive region of the world.

Since the differences between dry and wet phases in the model are–in line with the mean trend–not very large regarding the annual mean climate change, we cannot infer whether such

events were the main cause for the settlement crisis of the third millennium BCE/5 ka BP. The VP is located in an arid to semi-arid region with limited moisture supply. Therefore, even small changes could play a decisive role. MPI-ESM shows that towards the Late-Holocene, the number of drought years increases, and from 4 ka BP on the duration of periods with little precipitation expands. This suggests an increase in moisture stress for people in the VP towards and during the Late Holocene.

The general spatio-temporal change in rainfall in WCA over the last 8000 years is more homogenous in the model than in the pollen-based reconstructions, but is similar in both dataset. The 1st and 2nd principle components (PC) of the rainfall change resemble the Holocene precipitation trend in monsoon influenced regions (PC1) and the assumed Westerly wind response (PC2) and reveal that both signals are part of the precipitation change in WCA during the Holocene. Which mode dominates at certain sites and the intensity of the loadings may also be controlled by the local environmental conditions such as the topography that substantially affects the regional Westerly wind dynamics and the precipitation in WCA. It furthermore depends on the seasonality of precipitation and its temporal changes as rainfall seasonality determines how effectively orography or changes in the monsoon systems affect the precipitation pattern.

To conclude, our results indicate that climate may have played a role in settlement dynamics on the VP. Given the small absolute changes in moisture availability averaged over the year, it is, however, questionable whether climate could have been the primary driver of settlement growth, crisis, or abandonment. In fact, it would be unrealistic to assume one single or main cause for complex socio-economic changes. Other factors, such as shifting of trade routes or culturally/socially motivated changes in settlement patterns and forms of living could have been partially responsible for these changes. However, seasonality shifts may have played an as yet under-investigated role in these processes. Local climatic events have no spatially uniform manifestation. This underscores that misleading conclusions can easily result from comparing archaeological observations with proxy-based reconstructions that are not directly located in the archaeological survey regions. Clearly more high-resolution records are needed to conclusively address the question of whether changes in settlement density on the VP and the CIP were partly caused by climatic changes.

## Supporting information

**S1 Fig. Significant differences in monthly mean precipitation [mm/mon] between 7ka BP and PI, simulated by MPI-ESM.** Please note, that all differences relate to the calendar in the model, i.e. modern calendar.
(TIF)

**S2 Fig. Significant differences in monthly mean temperature [˚C] between 7ka BP and PI, simulated by MPI-ESM.** Please note, that all differences relate to the calendar in the model, i.e. modern calendar.
(TIF)

**S3 Fig. Significant differences in monthly mean soil moisture between 7ka BP and PI [mm], simulated by MPI-ESM.** Please note, that all differences relate to the calendar in the model, i.e. modern calendar.
(TIF)

**S1 Table. Archaeological periods and timeframes used in this study based on Bayesian age model median values calculated by Pollard et al. (2015); gaps between periods represent times of unknown classification; BP is commonly referenced to year 1950 (14C-datings,**

**etc.); MPI-ESM runs until 1850 CE (= Pre-Industrial "PI"), which means it ends exactly at 100 a BP.**
(XLSX)

## Acknowledgments

This study contributes to the project "Mobile villages and dynamic landscapes: the Varamin Plain from the late 5th to the early 3rd mill. BCE" within the priority programme 2176 "The Iranian Highlands: Resilience and Integration of Premodern Societies" of the German Research Foundation.

We thank U. Herzschuh for her support in supervision of Thomas Böhmer. We are grateful to L. Jungandreas for sharing the ICON simulation.

We thank Martin Claussen for his comments on an earlier version of this manuscript and two anonymous reviewers for their constructive comments.

## Author Contributions

**Conceptualization:** Fabian Kirsten, Anne Dallmeyer, Robert Busch, Brigitta Schütt.

**Formal analysis:** Fabian Kirsten, Anne Dallmeyer, Robert Busch.

**Funding acquisition:** Reinhard Bernbeck, Susan Pollock, Brigitta Schütt.

**Investigation:** Thomas Böhmer, Morteza Hessari.

**Methodology:** Fabian Kirsten, Anne Dallmeyer, Robert Busch.

**Project administration:** Reinhard Bernbeck, Susan Pollock.

**Resources:** Reinhard Bernbeck, Morteza Hessari, Susan Pollock.

**Supervision:** Reinhard Bernbeck, Susan Pollock, Brigitta Schütt.

**Visualization:** Fabian Kirsten, Anne Dallmeyer, Robert Busch.

**Writing – original draft:** Fabian Kirsten, Anne Dallmeyer, Thomas Böhmer.

**Writing – review & editing:** Fabian Kirsten, Anne Dallmeyer, Reinhard Bernbeck, Thomas Böhmer, Robert Busch, Susan Pollock, Brigitta Schütt.

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
