## [Decision Letter · Decision Letter 0]

28 Jun 2023

PONE-D-23-14544Were climatic forcings the main driver for Mid-Holocene changes in settlement dynamics on the Varamin Plain (Central Iranian Plateau)?PLOS ONE

Dear Dr. Kirsten,

Thank you for submitting your manuscript to PLOS ONE. After careful consideration, we feel that it has merit but does not fully meet PLOS ONE’s publication criteria as it currently stands. Therefore, we invite you to submit a revised version of the manuscript that addresses the points raised during the review process.

The two reviews appreciated your manuscript, specifically your approach and interpretation of the human-climate nexus along the Iranian plateau, but they also highlight several moderate to major points that need to be discussed before I can accept the paper for publication. Please, when revising you manuscript take care of all comments from the reviewers and especially from those related to the model you applied. Moreover, consider also (if possible) to shorten a bit your manuscript.I also appreciated your manuscript because I have experience in nearby areas (Iraqi Kurdistan) on similar topics and along with several co-authors I discussed the importance of climatic reconstruction in tracing the Holocene cultural dynamic of the western and central Asia in a recently published paper (Sc. Rep 2023). Please consider my potential conflict of interest in suggesting this manuscript, but try to have a look; in this paper we have a bit different perspective on the influence of climate on humans and a scientific discussion on this topic is of great interest. i also suggest to consider the review by Palmisano et al (2021) which also greatly contribute on the debate.This is a really hot topic and I am will be happy to consider soon the revised version of the manuscript for publication.

We look forward to receiving your revised manuscript.

Kind regards,

Andrea Zerboni, Ph.D.

Academic Editor

PLOS ONE

2. In your manuscript, please provide additional information regarding the specimens used in your study. Ensure that you have reported human remain specimen numbers and complete repository information, including museum name and geographic location.

For more information on PLOS ONE's requirements for paleontology and archeology research, see https://journals.plos.org/plosone/s/submission-guidelines#loc-paleontology-and-archaeology-research.

4. We note that Figures 1, 2, 4, 6, 7, 9, 10, 11, and 12 in your submission contain [map/satellite] images which may be copyrighted. All PLOS content is published under the Creative Commons Attribution License (CC BY 4.0), which means that the manuscript, images, and Supporting Information files will be freely available online, and any third party is permitted to access, download, copy, distribute, and use these materials in any way, even commercially, with proper attribution. For these reasons, we cannot publish previously copyrighted maps or satellite images created using proprietary data, such as Google software (Google Maps, Street View, and Earth). For more information, see our copyright guidelines: http://journals.plos.org/plosone/s/licenses-and-copyright.

a. You may seek permission from the original copyright holder of Figures 1, 2, 4, 6, 7, 9, 10, 11, and 12 to publish the content specifically under the CC BY 4.0 license. 

Reviewers' comments:

Reviewer's Responses to Questions

**Comments to the Author**

1. Is the manuscript technically sound, and do the data support the conclusions?

Reviewer #1: Yes

Reviewer #2: Yes

2. Has the statistical analysis been performed appropriately and rigorously? 

Reviewer #1: I Don't Know

Reviewer #2: Yes

3. Have the authors made all data underlying the findings in their manuscript fully available?

Reviewer #1: Yes

Reviewer #2: Yes

4. Is the manuscript presented in an intelligible fashion and written in standard English?

Reviewer #1: Yes

Reviewer #2: Yes

5. Review Comments to the Author

Reviewer #1: In the article, “Were climatic forcings the main driver for Mid-Holocene changes in settlement dynamics on the Varamin Plain (Central Iranian Plateau)?”, Kirsten et al., present the results of a range of climate models and compare these with palaeoclimate records to examine past changes in precipitation, moisture transport, temperature and more. They then compare this evidence to existing data to explore impacts on water resources and settlement dynamics. I am not an expert in climate modelling, but overall this manuscript shows great promise with in-depth synthesis and analysis of data/findings from multiple disciplines. I really enjoyed reading the manuscript and think the conclusions are very important. I will be citing and using this article extensively once published! Having said that, the text itself has some significant issues (please see the attached document), mostly related to certain sections being too long whilst others needed further explanation. I would therefore recommend the manuscript for publication after major revisions and would like to re-review the manuscript once these are completed.

Reviewer #2: Reviewer comments for ms PONE-D-23-14544 “Were climatic forcings the main driver for Mid-Holocene changes in settlement dynamics on the Varamin Plain (Central Iranian Plateau)?”

The ms from Kirsten et al. investigate the relationship between Holocene climatic changes and settlement patterns on the Central Iranian Plateau (CIP), and particularly on the Varanin Plain alluvial fan. The analyses is done by combining a transient comprehensive Earth System Model simulation (8 ka BP to pre-industrial, 200Km2 grid), a high-resolution (40 km2 grid) regional snapshot simulation and a synthesis of pollen-based climate reconstructions. Results are compared with proxy records from the wider Western-central Asia area (mostly speleothems). The ms examines the situation at 7 ka (when archaeological data show an increase in settlement density of the CIP/VP) compared to pre-industrial/modern time; and discusses both the general trend and the short-time changes highlighted by the simulation during the Middle-Late Holocene, in the context of the available paleoclimatic framework. Atmospheric patterns responsible for the simulated variability and the spatial heterogeneity of both simulation and proxy-based reconstruction are discussed.

The paper is of interest and the simulation results fit well with the available paleoclimatic context, suggesting that changes in the climate (mostly in the seasonality of the precipitation causing differences in water availability during spring-early summer) may have had an influence on settlement dynamics in the CIP/VP. Discrepancies between data and model are accurately reported and the model limitations (especially regarding the ability to capture the spatial heterogeneity of the climate due to the complex topography) are accounted for. The ms is well written and almost completely clear (also to non-modelling specialist), though a bit too long and with some repetitions; figures are of good quality. Overall, the ms can be published in Plos-One after moderate changes regarding two general points and several details (reported in the order as they appear in the ms)

General points:

1) PALEODATA-MODEL COMPARISON (and comparison with the Archaeological Dataset):

-I have some doubts on the use of the pollen-based LegacyClimate 2.0 dataset as a proper term of comparison/test of the model. I understand that it could be a nice synthesis to summarize global, general trends, but I’m not convinced that it can be used at this level of detail. Looking at the original publications it seems that the coverage of the study area (meaning CWA in its whole) is really poor. Moreover, pollen assemblages during the Middle-late Holocene are strongly influenced by anthropic activities, especially in CWA, making any reasoning about climatic influences on human societies a bit circular (i.e. is not possible to determine whether the pollen assemblages is modified by climate or by humans). Specifically, the harmonization to genus level made impossible to check the appearance of specific taxa regarded as strong indicators of anthropic modification to the local vegetation. This, in my opinion, made the discussion about the EOFs a bit speculative. I would consider removing it and rely instead on some recent syntheses (see e.g. Palmisano et al. 2021 Holocene regional population dynamics and climatic trends in the Near East: A first comparison using archaeo-demographic proxies. Quaternary Science Reviews, 252, 106739; and references therein). I understand the comparison will be more qualitative, but in my opinion also more reliable.

-I do not find a proper comparison between the model output and the archaeological dataset. The general settlement dynamics are discussed in the intro, largely repeated in section 2.3 and recalled in a qualitative fashion in comparison with the model results. This is a fair approach, but it make section 3.1 a bit useless, as the archeological data are compared only qualitatively, and not quantitatively

2) PAPER STRUCTURE AND LENGTH:

-The paper is too long and a bit repetitive. Several sections can be reduced and merged.

As example, Section 4.1.2 and its subsections can be easily reduced to few sentences, just saying that the model does support/does not support the different hypothesis. Similarly, the first part of section 4.2 is a repetition from the introduction, and also e.g. lines 791-797 are a repetition from the introduction and section 4.2. Also, the summary in the last section (5) is too long and detailed, you should consider that the reader arrives here straight after reading the text, so it sound like a long repetition. Other examples of repetition regards the archaeological sections (see above)

Minor points:

line 178: please define Qanats in English

line 263 please be consistent, use Ka BP and millennia BCE, I guess there is a typo here (2.8-2.2 ka BCE)

line 306 “Theis” should be a typo

line 347: I would say verying temporal resolution and related uncertainties

Figures

Fig.1 the different areas quoted in lines 125-133 (VP-CIP-WCA) should be reported on fig. 1

Fig. 5 is quoted before of Fig. 4 (line 303). Alo, given that the two models have a different reference periods (PI for MPI-ESM and modern climate for ICON-NWP) I would show also the comparison between PI and Modern, to check whether differences are significant or not.

Fig. 4 should be quoted in line 342

References:

I guess ref 10 and 11 are not the right ones here as they are quite old and both deals with long but relatively low resolution records from Soreq Cave speleothems. I would suggest to have a look to some recent compilations for the region (e.g. –see also general comments above)

Ref 30 is only submitted, it cannot be quoted in the ref list until it is at least accepted

6. PLOS authors have the option to publish the peer review history of their article (what does this mean?). If published, this will include your full peer review and any attached files.

Reviewer #1: No

Reviewer #2: No

---

## [Author Response · Author response to Decision Letter 0]

13 Jul 2023

Dear Dr Zerboni,

Thank you very much for your invitation to revise our manuscript “Were climatic forcings the main driver for Mid-Holocene changes in settlement dynamics on the Varamin Plain (Central Iranian Plateau)?”. We appreciate very much the suggestions and comments by the two reviewers, and we have revised our manuscript accordingly. In particular, we have substantially shorten the manuscript (Section 2.1, 4.1, 4.2 and the summary) and included the background on why settlement datasets are useful for comparison with climate data in the Introduction. We think that this has significantly improved the manuscript. We furthermore added a short comment on the possibility of a human impact on the pollen-based reconstructions in the method sections. We checked the abundance of taxa typically for grazing landscapes as indicator for human influence (via intense animal husbandry) on the pollen records. The abundances are low for most sites during the mid-Holocene, which is the main period of our analysis. We therefore assume a very weak anthropogenic impact on the climate reconstructions (see detailed answer to the reviewer). 

All changes we have made on the text are marked with the “track changes” option in the document. We have answered all requests by the reviewers in two separate files. 

Based on comments by Reviewer 1, we have chosen the following new keywords: 

Earth System Model, climate reconstruction; settlement dynamics, Mid-Holocene climate; Central Iran

We have ensured the permission to provide all datasets used for analysis and plotting in an open-access archive. Therefore we would like to change the data availability statement in our manuscript to:

“The speleothem-based climate reconstructions can be downloaded from the NOAA

Database (https://www.ncei.noaa.gov/access/paleo-search/). 

All other datasets and analyse scripts used for the analysis and plots in this study are deposited in the MPG publication repository: https://hdl.handle.net/21.11116/0000-000D-14BA-B and

can be downloaded free of charge.

In addition, a preliminary version of the pollen-based climate reconstruction (LegacyClimate 1.0) for the Northern hemispheric extratropics is provided as open-access data on PANGAEA

(https://doi.pangaea.de/10.1594/PANGAEA.930512). 

Further variables of the high-resolution snapshot simulation can be downloaded from the long term archive of the German Climate Computing Center (DKRZ), accredited as regular member of the World Data System (https://doi.org/10.26050/WDCC/ICON-NWP_mH_pd).”

We will provide the final archive as soon as our manuscript is invited for publication. 

The comments by the editorial office regarding journal requirements (formatting, data availability and permissions for figures) have been taken into account. With respect to all figures mentioned, we used a feature layer named “Global_Ocean_Country_Masks” (ID: 91) in ArcGIS Online, which is freely available for users of ArcGISPro. To our knowledge, no further permission or license is required to use this dataset in publications. We have added a reference to this dataset to all figures in which it was used.

As Fabian Kirsten will not be available via email between the 14th of July and the 13th of August, please contact Anne Dallmeyer with any questions regarding this revision during this time period. Unfortunately, it is not possible to include a second corresponding author in the EditorialManager.

Sincerely yours,

 Fabian Kirsten and Anne Dallmeyer

---

## [Decision Letter · Decision Letter 1]

3 Aug 2023

Were climatic forcings the main driver for Mid-Holocene changes in settlement dynamics on the Varamin Plain (Central Iranian Plateau)?

PONE-D-23-14544R1

Dear Dr. Kirsten,

We’re pleased to inform you that your manuscript has been judged scientifically suitable for publication and will be formally accepted for publication once it meets all outstanding technical requirements.

Kind regards,

Andrea Zerboni, Ph.D.

Academic Editor

PLOS ONE

Additional Editor Comments (optional):

Reviewers' comments:

Reviewer's Responses to Questions

**Comments to the Author**

1. If the authors have adequately addressed your comments raised in a previous round of review and you feel that this manuscript is now acceptable for publication, you may indicate that here to bypass the “Comments to the Author” section, enter your conflict of interest statement in the “Confidential to Editor” section, and submit your "Accept" recommendation.

Reviewer #1: All comments have been addressed

2. Is the manuscript technically sound, and do the data support the conclusions?

Reviewer #1: Yes

3. Has the statistical analysis been performed appropriately and rigorously? 

Reviewer #1: I Don't Know

4. Have the authors made all data underlying the findings in their manuscript fully available?

Reviewer #1: Yes

5. Is the manuscript presented in an intelligible fashion and written in standard English?

Reviewer #1: Yes

6. Review Comments to the Author

Reviewer #1: Thank you very much for your detailed response. I am happy that you have addressed all my comments, or justified that the suggested changes were unnecessary. The paper is much improved, especially by the additions to 4.2.

7. PLOS authors have the option to publish the peer review history of their article (what does this mean?). If published, this will include your full peer review and any attached files.

Reviewer #1: No

---

## [Editor Report · Acceptance letter]

30 Aug 2023

PONE-D-23-14544R1 

Were climatic forcings the main driver for Mid-Holocene changes in settlement dynamics on the Varamin Plain (Central Iranian Plateau)? 

Dear Dr. Kirsten:

I'm pleased to inform you that your manuscript has been deemed suitable for publication in PLOS ONE. Congratulations! Your manuscript is now with our production department. 

Kind regards, 

on behalf of

Prof. Andrea Zerboni 

Academic Editor

PLOS ONE